# Efficient Multimodal Planning Agent for Visual Question-Answering

## Abstract

Visual Question-Answering (VQA) is a challenging multimodal task that requires integrating visual and textual information to generate accurate responses. While multimodal Retrieval-Augmented Generation (mRAG) has shown promise in enhancing VQA systems by providing more evidence on both image and text sides, the default procedure that addresses VQA queries, especially the knowledge-intensive ones, often relies on multi-stage pipelines of mRAG with inherent dependencies. To mitigate the inefficiency limitations while maintaining VQA task performance, this paper proposes a method that trains a multimodal planning agent, dynamically decomposing the mRAG pipeline to solve the VQA task. Our method optimizes the trade-off between efficiency and effectiveness by training the agent to intelligently determine the necessity of each mRAG step. In our experiments, the agent can help reduce redundant computations, cutting search time by over 60% compared to existing methods and decreasing costly image retrieval calls. Meanwhile, experiments demonstrate that our method outperforms all baselines, including a carefully designed prompt-based method, on average over six various datasets. Code will be released at `https://github.com`

## 1 Introduction

Visual Question-Answering (VQA) is a fundamental task in multimodal artificial intelligence that requires the ability to understand and integrate both visual and textual information to produce correct answers (Cheng et al., 2025; Lu et al., 2024b; Bai et al., 2025b). Recent studies have demonstrated advancements in this area, focusing on improving model performance across different types of VQA queries. These include knowledge-intensive questions that require external factual information (Wen et al., 2024) as well as dynamic queries where answers may change over time (Li et al., 2025). Various methods have been studied to integrate multimodal Retrieval-Augmented Generation (mRAG) to better solve various types of VQA queries. These studies typically enhance the capabilities of models by incorporating retrieved evidence from both visual and textual sources (Chen et al., 2025; Xue et al., 2024; Xenos et al., 2023), and further advance Multimodal Large Language Models' (MLLMs) potential in real-world applications.

However, a key limitation constrains the practical efficiency and scalability of existing mRAG systems. Current implementations typically employ rigid, multi-stage pipelines, possibly involving image grounding (Adjali et al., 2024), image retrieval (Jian et al., 2024), and query rewriting (potentially using retrieved contexts) (Zhu et al., 2024; Liu & Mozafari, 2024), followed by text passage retrieval (Li et al., 2025; Adjali et al., 2024). Besides, these steps may also exhibit inherent potential dependencies. For instance, effective query rewriting often necessitates prior image retrieval to provide additional information about the image content, while text retrieval has a critical dependency on query rewriting (Ma et al., 2023). These static workflows are inefficient and remain data-agnostic, often lacking dynamic selection mechanisms between processing stages. Also, redundant retrieval steps introduce overly long input length. Consequently, valuable computational resources are expended even when the original input query might be sufficiently answered using readily available cues alone, or when certain steps provide marginal benefit for a relatively simple query.

To mitigate inefficiency without compromising performance, this paper introduces a multimodal planning agent designed to enhance the efficiency of mRAG pipelines in VQA tasks by dynamically adapting to diverse queries. The agent takes necessary steps given a VQA query on a workflow

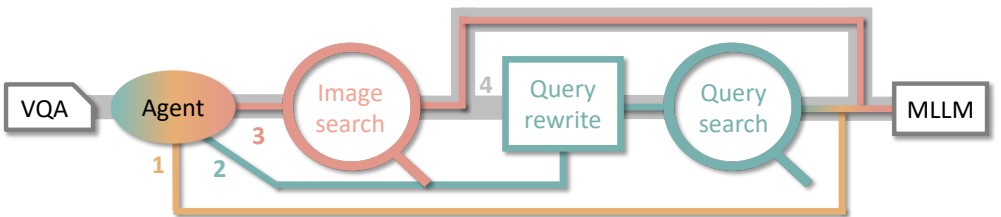

Figure 1: Workflow of our agent on solving VQA with dynamic mRAG strategies. The agent selects a sub-path based on different VQA inputs, which may require image search, query search, neither, or both.

as illustrated in Fig. 1. In general, facing various VQA queries at test time, the agent optimizes computational resource allocation by intelligently omitting redundant operations. Specifically, for queries necessitating external knowledge or specialized tools, the agent strategically decomposes the mRAG workflow, selectively executing only those components essential for generating accurate responses (path 2 or 3), thereby departing from rigid pipeline architectures (path 4). In addition, for simpler queries resolvable via the model's intrinsic capabilities, the agent learns to bypass extraneous processing steps entirely (path 1).

Through experiments across six diverse VQA datasets, we demonstrate the efficiency and effectiveness of our method. The agent helps achieve substantial gains in inference efficiency compared to both the complete mRAG setting and a designed prompt-based method *OmniSearch* (Li et al., 2025). Notably, compared to *OmniSearch*, we reduce the search time by 60%+ on average, and significantly lower the number of expensive image-search calls. Furthermore, this significant efficiency improvement is attained while enhancing or maintaining the VQA task performance on average over six datasets compared to the default complete mRAG setting and all other baseline methods.

To sum up, our contributions are as follows

1. The paper proposes a multimodal planning agent that dynamically optimizes mRAG pipelines while maintaining VQA performance with higher efficiency.
2. Experimental results across diverse VQA datasets show that the agent significantly reduces search time (60%+ compared to a designed prompt-based method) and costly retrieval operations compared to baseline methods. In addition, we obtain improved performance on average over six test datasets.

## 2 RELATED WORK

Recent advances in MLLMs have enabled more sophisticated agent-based systems for multimodal tasks like the VQA task (Xie et al., 2024; Gao et al., 2023; Jiang et al., 2024). These agents often integrate RAG mechanisms to enhance reasoning by dynamically retrieving and incorporating external knowledge from both visual and textual modalities (Song et al., 2025). A common approach involves multi-stage pipelines where agents sequentially perform operations such as image retrieval, query refinement (Zhu et al., 2024), and text retrieval before generating an answer. While this paradigm improves accuracy by leveraging external evidence, it introduces inefficiencies due to rigid step-by-step execution, where later stages depend on the outputs of earlier ones. Besides, it may lead to improper use of tools (e.g., unnecessary retrieval) and the incorporation of excessively long contexts into the input.

### 2.1 EXISTING MULTIMODAL PLANNING AGENT FOR VQA

Recent work has explored prompt-based methods to optimize mRAG pipelines. These approaches typically depend on the inherent capabilities of the underlying pretrained models and complicated prompt engineering. Within the prompt-based paradigm, multimodal models are prompted to select actions from a predefined action space and perform these selected actions on external tools, such as retrieval systems. Subsequently, the outputs from these tools, in conjunction with the original

input, are iteratively fed back into the model in a recurrent manner, enabling continuous reasoning and interaction. Li et al. (2025) proposes *OmniSearch*, which emulates human behavior in inference stage and dynamically decomposes complex multimodal questions into sub-question chains with retrieval action via designed prompts. Such methods primarily rely on the model's strong capability in following instructions, as the output generated from tool invocation must adhere to a relatively strict format, such as JSON. Any deviation or error in the output format will lead to the failure of the entire approach. A model that lacks reliable instruction-following capabilities becomes fundamentally uncontrollable in VQA settings with mRAG.

Besides prompt-based methods, Chen et al. (2025) introduced an automated process for detecting the "*knowledge boundary*" by fine-tuning an MLLM based on automatically sampled data. The *knowledge boundary* stands for a concept of dividing line between what the model knows and what the model does not know. The fine-tuning better guarantees the instruction-following ability. However, classifying a VQA query as inside or outside the knowledge boundary does not, by itself, provide a mechanism for handling insufficient knowledge. Specifically, it cannot determine whether external textual or visual information should be retrieved to ensure an accurate response.

In this work, we extend Chen et al. (2025)'s method by endowing the model with the ability to dynamically select necessary components in a predefined workflow like an agent, rather than merely detecting *knowledge boundaries*. This enhancement significantly improves adaptability in open-domain VQA scenarios at inference time across various questions. By integrating actionable decision-making into the retrieval process, our method advances beyond static knowledge assessment toward intelligent, adaptive multimodal planning.

## 3 METHOD

We propose a method that initially performs data annotation via VQA query decomposition, followed by fine-tuning of an MLLM agent. This section begins by establishing the requisite mathematical notations. Subsequently, we elaborate on the automated annotation process and detail the procedures for agent training and inference. The fine-tuned agent operates in alignment with the workflow illustrated in Fig. 1.

### 3.1 NOTATIONS

Let $\boldsymbol{q} = (\boldsymbol{i}, \boldsymbol{t})$ represent a VQA query composed of image input $\boldsymbol{i}$ and a textual question component $\boldsymbol{t}$, and let $\boldsymbol{a}$ denote the corresponding ground truth answer. In general VQA tasks, the original textual query $\boldsymbol{t}$ may require reformulation into an optimized query $\boldsymbol{q}_g$ (henceforth may be referred to as gold query) to more accurately characterize the information needs expressed in $\boldsymbol{q}$. For example, in a situation where the query $\boldsymbol{q}$ asks *"When did this sorority established a chapter at American University"*, the gold query $\boldsymbol{q}_g$ should be *"When was ⟨Name⟩ established at American University?"*, and ⟨Name⟩ refers to the actual sorority name in the image. Let $\boldsymbol{k_i}$ denote the set of multimodal contextual elements retrieved using visual input $\boldsymbol{i}$. Let $\boldsymbol{k_t}$ denote textual contexts obtained through the optimized query $\boldsymbol{q}_g$. For an MLLM $M_\theta$ parameterized by $\theta$, the answer generation process, when relying solely on the MLLM, can be formally characterized by:

$$\boldsymbol{y}_n = M_\theta(\boldsymbol{y}|\boldsymbol{q}) \tag{1}$$

Generation with retrieval information from image retrieval, text retrieval, and both sides can be formulated as:

$$\boldsymbol{y}_i = M_\theta(\boldsymbol{y}|\boldsymbol{q}, \boldsymbol{k}_i); \quad \boldsymbol{y}_t = M_\theta(\boldsymbol{y}|\boldsymbol{q}, \boldsymbol{k}_t); \quad \boldsymbol{y}_{i,t} = M_\theta(\boldsymbol{y}|\boldsymbol{q}, \boldsymbol{k}_i, \boldsymbol{k}_t) \tag{2}$$

### 3.2 AGENT TRAINING DATA

**Visual Query Decomposition** To construct the training data of the agent, given one $(\boldsymbol{q}, \boldsymbol{a})$ pair as an example, we further expand it into two derived queries $\boldsymbol{q}_i$ and $\boldsymbol{q}_g$. $\boldsymbol{q}_i$ refers to a *image query* that queries what is in the image (e.g., asking the entity in the image). $\boldsymbol{q}_g$ refers to a gold query that combines $\boldsymbol{i}$ and $\boldsymbol{t}$ and more comprehensively describes the required information. Accordingly, we also need the corresponding gold answer for image query $\boldsymbol{q}_i$ and gold query $\boldsymbol{q}_g$. The answer to $\boldsymbol{q}_i$ is basically the image entity or a detailed description of the image $\boldsymbol{i}$, and the answer to $\boldsymbol{q}_g$ is $\boldsymbol{a}$.

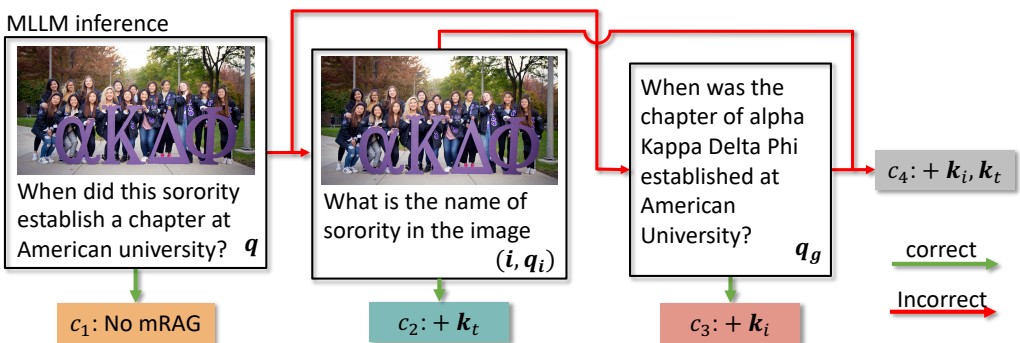

Figure 2: Proposed data annotation method.

This decomposition necessitates the generation of three new components: image query $q_i$, image entity $a_i$, and gold query $q_g$. Due to the large size of the training set, we adopt a strong MLLM to annotate these three components. $q_i$ and $a_i$ are generated conditioned on the original question $q$ and answer $a$. The gold query is re-written conditioned on $q$ and $a$. Notably, $q_g$ is used during both training and inference. Its generation at inference time requires an alternative annotation procedure that does not rely on the availability of the gold answer $a$ and is defined later. Detailed prompts of these procedures are shown in Sec A.1.

**Data Annotation** Based on the three queries $q$, $q_i$ and $q_g$ and their corresponding answers, we consider partitioning $q$ into four categories $c_{1-4}$, as illustraed in Fig. 2:

- $c_1$. No mRAG is needed, if $M_\theta(y|q)$ is correct.
- $c_2$. More contexts $k_t$ related to the textual input are needed, if $M_\theta(y|q)$ is incorrect but $M_\theta(y|i, q_i)$ is correct.
- $c_3$. More contexts $k_i$ related to the visual input are needed, if $M_\theta(y|q)$ is incorrect but $M_\theta(y|q_g)$ is correct.
- $c_4$. Both $k_i$ and $k_t$ are needed, if all $M_\theta(y|q)$, $M_\theta(y|q_i)$ and $M_\theta(y|q_g)$ are incorrect.

Strictly speaking, the model may incorrectly answer $q$ while correctly answering $q_i$ and $q_g$. These cases were rare in our experiments, and we excluded them from training as they conflict with conventional logic (if a model successfully recognizes the image and correctly answers the gold query, it should be sufficient to answer the original query).

## 3.3 Agent Training and Inference

**Training** For VQA query $q = (i, t)$, with its category label $c$ properly annotated according to the previous section, we can fine-tune[1] the MLLM $M_\theta$ to operate like an agent. $\theta$ is optimized w.r.t. minimizing

$$J(\theta) = -\sum_{q \in \mathcal{D}} \log P_\theta(c|q, T) \tag{3}$$

where $P_\theta(a|b)$ stands for the probability model parameterized by $\theta$ predicting on $a$ given input $b$. Denote the optimized parameters by $\theta^*$. $T$ refers to prompts towards predicting category $c$. Detailed form of $T$ is shown in Sec. A.2. $\mathcal{D}$ stands for the training set.

**Inference** With optimized $\theta^*$, the agent selects one of the four categories defined in Sec. 3.2 with prompt $T$, operating as an agent adhering to the workflow depicted in Fig. 1. It is worth noting that gold queries $q_g$ are usually missing at inference time. Thus, it becomes difficult to retrieve $k_t$ if the agent predicts category $c_2$ or $c_4$. Here we provide a specific formulation of $q_g$ at inference time and the following inference process with task model $M_\phi$ ($\phi$ can be either $\theta$ or other open/closed-source models). Given a VQA query $q$ and prompt $T$, if the agent predicts to adopt:

---

[1]This paper considers the setting where the agent model to be trained is the same as the one used for data annotation, thereby we can choose open-source models.

$c_1$. No mRAG, generating a gold query is unnecessary. Downstream models $M_\phi$ directly run inference on $\boldsymbol{q}$: $M_\phi(\boldsymbol{y}|\boldsymbol{q})$.

$c_2$. More contexts $\boldsymbol{k}_t$, the gold query $\boldsymbol{q}_g$ is rewritten given the original VQA query $\boldsymbol{q}$ by an MLLM[2]. Contexts $\boldsymbol{k}_t$ are retrieved using $\boldsymbol{q}_g$. The inference is $M_\phi(\boldsymbol{y}|\boldsymbol{q}, \boldsymbol{k}_t)$.

$c_3$. More contexts $\boldsymbol{k}_i$, generating a gold query is unnecessary, and only $\boldsymbol{k}_i$ will be supplemented in the following inference process: $M_\phi(\boldsymbol{y}|\boldsymbol{q}, \boldsymbol{k}_i)$.

$c_4$. Both $\boldsymbol{k}_t$ and $\boldsymbol{k}_i$, $\boldsymbol{k}_i$ will be first retrieved using image $\boldsymbol{i}$. Following that, $\boldsymbol{k}_i$ and the original VQA query $\boldsymbol{q}$ are used to rewrite[3] the gold query $\boldsymbol{q}_g$. The inference is $M_\phi(\boldsymbol{y}|\boldsymbol{q}, \boldsymbol{k}_i, \boldsymbol{k}_t)$

## 4 EXPERIMENT

### 4.1 SETUP

#### 4.1.1 TRAINING SETTING

When constructing the training set according to Sec. 3.2, we experiment with Qwen2.5-VL-7B-Inst (Bai et al., 2025a) as $M_\theta$. Qwen-Max (Team, 2024) is prompted to evaluate the correctness of the responses. Qwen2.5-VL-72B (Bai et al., 2025a) is used to perform query rewriting and generate gold query $\boldsymbol{q}_g$, image query $\boldsymbol{q}_i$ and the answer $\boldsymbol{a}_i$ to $\boldsymbol{q}_i$. We apply LoRA (Hu et al., 2021) and full fine-tuning to train the agent and find that LoRA with rank 32 works fairly well compared to full fine-tuning. In the subsequent sections, we default to showing the results of training using LoRA. The result of full fine-tuning is also reported in Sec. A.4. Refer to Sec. A.3 for detailed hyperparameter and training cost.

#### 4.1.2 TRAINING DATA

We adopt InfoSeek (Chen et al., 2023) and VQAv2.0 (Goyal et al., 2017), following Chen et al. (2025), as source datasets to construct the training set. Additionally, we introduce WanWu, a Chinese VQA dataset covering news figures, events, animals, and plant-related questions. WanWu is also incorporated as a source training set. We report detailed statistics of training data in Table 1.

#### 4.1.3 TEST DATA

The proposed agent is designed to address diverse types of VQA queries, including knowledge-intensive ones, queries with static or dynamic knowledge, etc. To validate its performance, we evaluate our method across the following six test datasets with varying characteristics. All test datasets are completely isolated from the training sets. The specific quantities and properties of each dataset are summarized in Table 2.

**Dyn-VQA (ch/en)** is introduced by Li et al. (2025) with Chinese and English versions. It comprises three distinct question categories: (1) questions with time-sensitive answers, (2) questions demanding multi-modal knowledge, and (3) multi-hop reasoning questions. Due to its complexity, this dataset serves as a challenging benchmark in our evaluation. We generate the gold query $\boldsymbol{g}$ according to the procedure stated in Sec. 3.3 instead of the provided ones.

**Life VQA** is introduced by Chen et al. (2025), consisting of real-world visual questions curated from daily life scenarios, specifically targeting cases where existing MLLMs exhibit poorly.

**Private VQA** constitutes an internal collection encompassing diverse categories such as fauna, flora, architectural structures, and geographical settings. The intricate background compositions and frequent multi-object scenarios present in this dataset establish it as a significant benchmark for evaluating sophisticated visual comprehension and reasoning capabilities.

**NoCaps** (Agrawal et al., 2019) is built upon the Open Images dataset (Krasin et al., 2017), evaluates open-domain image captioning performance across diverse object categories and scene types. For our experiments, we utilize a randomly selected subset of 500 instances.

---

[2] Refer to Sec. A.1 for detailed prompt to obtain $\boldsymbol{q}_g$. In this paper, we employ a fixed query rewriting model across all experimental settings to ensure methodological consistency.

[3] Refer to Sec. A.1 for detailed prompt to obtain $\boldsymbol{q}_g$ when $\boldsymbol{k}_i$ is available.

| Training Data Source | Quantity |
|---|---|
| InfoSeek | 53,999 |
| VQAv2.0 | 53,180 |
| Wanwu | 66,076 |
| **Total (Raw)** | **173255** |
| *Final Training Set (by Category)* | |
| No mRAG | 30,000 |
| Image mRAG | 8806 |
| Query mRAG | 30,000 |
| Both mRAG | 30,000 |
| **Total (Final)** | **98806** |

Table 1: Statistics of the training dataset.

| Test Data | Quantity | mRAG Effect |
|---|---|---|
| Life VQA | 149 | High |
| Private VQA | 500 | Medium |
| Dyn-VQA ch | 737 | High |
| Dyn-VQA en | 715 | High |
| NoCaps | 500 | Low |
| Visual7W | 574 | Low |
| Mix | 600 | Mixed |

Table 2: Test data property illustration of quantity and whether mRAG is helpful.

**Visual7W**   (Zhu et al., 2016), derived from MS COCO images (Lin et al., 2014), presents question-answer pairs spanning seven fundamental interrogative types (who, what, when, where, how, why, and which). This benchmark comprehensively assesses both basic visual recognition and advanced contextual reasoning capabilities.

**Mix**   dataset contains 100 random samples from each source test dataset, combining their distinct features to simulate real-world conditions. The effect of applying mRAG on this dataset becomes mixed and unpredictable because it contains various types of VQA queries.

## 4.2 MAIN RESULTS

In this section, we present the result where we adopt Qwen2.5-VL-7B-Inst as the task model. Besides the officially released instructed version, we also experiment with Qwen2.5-VL-7B parameterized by $\theta^*$ (i.e., the fine-tuned agent itself is applied to VQA tasks). We present the results of more MLLMs serving as task models in Sec. 5.2.

We report the task performance and ratios of each retrieval type in Table 3. Scores shown in the table (except for the **%** ones) are LLM evaluation scores, ranging from 0 to 100. Higher scores refer to higher performances. We also report the performance evaluated with a static metric, token accuracy, in Sec. A.5. **%** columns represent the proportion of the agent's predictions corresponding to each mRAG type. For instance, the notation **%** $+k_i$ indicates the ratio of scenarios where the agent's decision exclusively adopts image retrieval.

First, the results in the **Mix** row, which considers all kinds of VQA queries and simulates a real situation, show that with the mRAG planning agent, our methods outperform all other baseline settings. Notably, while the $+k_{i,t}$ configuration establishes a remarkably strong baseline at the cost of computational efficiency, our proposed method consistently surpasses its performance both on the **Mix** dataset and in terms of unweighted average (**Avg.**) metrics.

Second, as shown by the **%** columns, our planning agent dynamically predicts the retrieval type regarding different datasets. For example, the agent predicts not to adopt mRAG (~60%) on Nocaps and Visual7W datasets, where the queries tend to be solvable using MLLM only. Also, compared to the **Prompt-based** baseline, where the model is overly confident in adopting mRAG, our agent performs better at utilizing image retrieval and text retrieval tools on other datasets.

Lastly, the result across the first four datasets reveals that one or more of $k_i$, $k_t$ and $k_{i,t}$ can significantly enhance performance on VQA tasks, indicating that these particular data types benefit more substantially from mRAG. Our method demonstrates that: (1) it achieves performance comparable to or even surpassing the optimal mRAG configuration. E.g., our method reaches 56.48 on Dyn-VQA (en) data while the $+k_{i,t}$ setting reaches 56.34; (2) it enables more efficient mRAG deployment by eliminating the need for simultaneous searches across both textual and visual content, e.g., we maintain the performance on Private VQA while keeping only 36.4% $+k_{i,t}$ mRAG.

## 5 ANALYSIS

In this section, we first compare our method with a designed prompt-based method, *OmniSearch* (Li et al., 2025). *OmniSearch* incorporates tools including image-to-image, text-to-text, and text-

| *Metric: LLM Eval.* | | No mRAG | $+k_i$ | $+k_t$ | $+k_{i,t}$ | Pt.-based | % No | % $+k_i$ | % $+k_t$ | % $+k_{i,t}$ | Ours | % No | % $+k_i$ | % $+k_t$ | % $+k_{i,t}$ |
|---|---|---|---|---|---|---|---|---|---|---|---|---|---|---|---|
| Life VQA | Qwen2.5 | 59.19 | 75.40 | 55.23 | 74.05 | 59.19 | 99.3 | 0.7 | 0.0 | 0.0 | 71.81 | 8.1 | 22.8 | 38.3 | 30.9 |
| | *-Agent | 57.85 | 70.91 | 49.66 | 70.74 | 57.85 | 99.3 | 0.7 | 0.0 | 0.0 | 67.56 | 8.1 | 22.8 | 38.3 | 30.9 |
| Private VQA | Qwen2.5 | 50.46 | 59.78 | 48.98 | 57.74 | 50.90 | 97.2 | 2.8 | 0.0 | 0.0 | 56.40 | 5.6 | 18.6 | 39.4 | 36.4 |
| | *-Agent | 50.42 | 55.30 | 46.31 | 55.24 | 50.44 | 97.2 | 2.8 | 0.0 | 0.0 | 54.86 | 5.6 | 18.6 | 39.4 | 36.4 |
| Dyn-VQA (ch) | Qwen2.5 | 43.73 | 47.12 | 50.80 | 57.58 | 44.45 | 80.1 | 19.9 | 0.0 | 0.0 | 55.51 | 1.6 | 13.0 | 56.0 | 29.3 |
| | *-Agent | 42.40 | 41.78 | 47.15 | 56.45 | 43.24 | 80.1 | 19.9 | 0.0 | 0.0 | 52.29 | 1.6 | 13.0 | 56.0 | 29.3 |
| Dyn-VQA (en) | Qwen2.5 | 49.53 | 50.10 | 52.39 | 56.34 | 49.04 | 69.1 | 30.3 | 0.6 | 0.0 | 56.48 | 14.1 | 3.4 | 62.2 | 20.3 |
| | *-Agent | 44.71 | 42.29 | 51.34 | 53.27 | 43.92 | 69.1 | 30.3 | 0.6 | 0.0 | 53.79 | 14.1 | 3.4 | 62.2 | 20.3 |
| Visual7W | Qwen2.5 | 75.72 | 70.88 | 67.42 | 65.24 | 75.43 | 97.6 | 2.4 | 0.0 | 0.0 | 71.38 | 60.1 | 1.4 | 30.8 | 7.7 |
| | *-Agent | 75.47 | 65.26 | 60.96 | 59.97 | 75.19 | 97.6 | 2.4 | 0.0 | 0.0 | 70.42 | 60.1 | 1.4 | 30.8 | 7.7 |
| NoCaps | Qwen2.5 | 80.44 | 77.30 | 80.70 | 76.60 | 80.30 | 98.6 | 0.4 | 0.0 | 1.0 | 80.36 | 58.8 | 0.0 | 40.4 | 0.8 |
| | *-Agent | 79.44 | 72.80 | 78.66 | 68.28 | 79.40 | 98.6 | 0.4 | 0.0 | 1.0 | 78.86 | 58.8 | 0.0 | 40.4 | 0.8 |
| Mix | Qwen2.5 | 58.81 | 62.79 | 58.51 | 64.41 | 58.68 | 89.0 | 10.8 | 0.0 | 0.2 | **64.93** | 24.8 | 9.5 | 43.3 | 22.3 |
| | *-Agent | 56.53 | 58.23 | 54.41 | 60.93 | 57.08 | 89.0 | 10.8 | 0.0 | 0.2 | **62.76** | 24.8 | 9.5 | 43.3 | 22.3 |
| Avg. | Qwen2.5 | 59.85 | 63.43 | 59.25 | 64.59 | 59.89 | 90.3 | 9.4 | 0.1 | 0.2 | **65.32** | 24.7 | 9.9 | 44.5 | 20.9 |
| | *-Agent | 58.38 | 58.06 | 55.68 | 60.66 | 58.34 | 90.3 | 9.4 | 0.1 | 0.2 | **62.96** | 24.7 | 9.9 | 44.5 | 20.9 |

Table 3: Main result on fine-tuned Qwen2.5-VL-7B serving as mRAG planning agent. **Qwen2.5** refers to the officially released Qwen2.5-VL-7B-Inst as the VQA solver and its fine-tuned version serving as the mRAG planning agent. **\*-Agent** stands for the result where the fine-tuned agent itself is also used to infer VQA queries. **No mRAG** refers to the setting where MLLM does not rely on any form of RAG. **Pt.-based** refers to the prompt-based baseline methods where the original Qwen2.5-VL-7B-Inst is prompted to output one of the mRAG types defined in Sec. 3.2. $+k_*$ columns stand for the performances when universally adopting the corresponding mRAG on all examples.

to-image search, and also supports multi-round conversations. We conduct a comparative analysis of our method and *OmniSearch* on several datasets we evaluated in the main result section. Subsequently, we investigate the transferability of our agent model by evaluating its performance when applied to other MLLMs. In the last subsection, we present an empirical analysis of the agent's training dynamics, examining both full fine-tuning and LoRA with rank 8 and 32.

## 5.1 COMPARE WITH *OmniSearch*

*OmniSearch* is a strong method with the capability to intelligently invoke tools for solving VQA tasks. This subsection presents a comparative analysis between our method and *OmniSearch*. The comparison encompasses first the performance on the VQA task, and second, the number of tool retrieval operations required, as well as the corresponding execution time when processing the same test set. Tools consist of image-to-image (**i2i**), text-to-text (**t2t**), and text-to-image (**t2i**) search. Search time is calculated by multiplying the average time for each searching tool by the count and taking the sum. Empirical measurements of the average processing duration for image-to-image, text-to-text, and text-to-image retrieval operations through our API endpoints yielded results of 6.4 seconds, 1.4 seconds, and 1.9 seconds, respectively. In our measurements, agent inference takes 1.65 s/sample on a single A100-SXM-80G GPU. Search time and agent inference latency comparison is shown in Table 4. Detailed components of searching time are shown in Fig 3.

The experimental results demonstrate that our method consistently achieves superior performance compared to *OmniSearch*, exhibiting an average reduction of 66.7% in search time during testing. Specifically, on the Dyn-VQA (en) dataset, our method reduces image-to-image search operations by 87.4% and text-to-text search operations by 69.8%, while simultaneously enhancing overall task performance. It is noteworthy that image-to-image search operations represent a significant bottleneck in vision-language agents, contributing substantially to increased latency. In our method, decreased retrieval frequency results in shorter input sequences, which subsequently reduces the computational burden during inference. Considering the agent's inference time, our method still reduces the time by 52%.

## 5.2 AGENT APPLYING TO MORE MLLMS

We also apply the fine-tuned agent model across diverse MLLMs: a same-scale 7B model (DeepSeek-VL-Chat; Lu et al. 2024a), two larger-scale variants from the same source (Qwen-VL-

| Base Model: GPT-4o | Search time ↓ OmniS. | Search time ↓ Ours | Agent infer. ↓ Ours | Sum ↓ Ours |
|---|---|---|---|---|
| Life. | 1110.8 | 656.2 | 245.9 | **902.1** |
| Private. | 4194.6 | 2290.6 | 825.0 | **3115.6** |
| Dyn (ch) | 6473.1 | 2876.0 | 1216.1 | **4092.1** |
| Dyn-(en) | 11449.4 | 1907.6 | 1179.8 | **3087.4** |
| Avg. | 5807.0 | 1932.6 | 866.7 | **2799.3** |

Table 4: Comparison to *OmniSearch*. **Search time** denotes the total search duration (seconds). **Agent infer.** denotes the planning agent's inference latency.

Figure 3: Comparison to OmniSearch with performance and detailed retrieval counts. The layered bars show the search counts of each **i2i**, **t2t**, and **t2i** type.

| Metric: LLM Eval. | | No mRAG | $+k_i$ | $+k_t$ | $+k_{i,t}$ | Pt.-based | % No | % $+k_i$ | % $+k_t$ | % $+k_{i,t}$ | Ours | % No | % $+k_i$ | % $+k_t$ | % $+k_{i,t}$ |
|---|---|---|---|---|---|---|---|---|---|---|---|---|---|---|---|
| **Life VQA** | DS-7B | 41.21 | 46.38 | 40.54 | 71.14 | 41.34 | 99.3 | 0.7 | 0.0 | 0.0 | 58.59 | 8.1 | 22.8 | 38.3 | 30.9 |
| | GPT-4o | 63.11 | 70.72 | 57.38 | 71.41 | 63.11 | 99.3 | 0.7 | 0.0 | 0.0 | 68.97 | 8.1 | 22.8 | 38.3 | 30.9 |
| | Q-Max | 59.33 | 68.81 | 53.42 | 71.07 | 59.19 | 99.3 | 0.7 | 0.0 | 0.0 | 69.37 | 8.1 | 22.8 | 38.3 | 30.9 |
| | Q-latest | 62.79 | 72.01 | 61.34 | 73.62 | 62.79 | 99.3 | 0.7 | 0.0 | 0.0 | 75.74 | 8.1 | 22.8 | 38.3 | 30.9 |
| **Private VQA** | DS-7B | 37.76 | 48.98 | 37.52 | 50.62 | 38.14 | 97.2 | 2.8 | 0.0 | 0.0 | 46.67 | 5.6 | 18.6 | 39.4 | 36.4 |
| | GPT-4o | 57.68 | 55.60 | 54.44 | 61.48 | 57.70 | 97.2 | 2.8 | 0.0 | 0.0 | 58.86 | 5.6 | 18.6 | 39.4 | 36.4 |
| | Q-Max | 51.80 | 57.33 | 49.04 | 57.44 | 52.44 | 97.2 | 2.8 | 0.0 | 0.0 | 56.28 | 5.6 | 18.6 | 39.4 | 36.4 |
| | Q-latest | 55.36 | 57.86 | 53.74 | 59.28 | 55.36 | 97.2 | 2.8 | 0.0 | 0.0 | 59.34 | 5.6 | 18.6 | 39.4 | 36.4 |
| **Dyn-VQA (ch)** | DS-7B | 35.17 | 35.83 | 46.01 | 55.41 | 35.21 | 80.1 | 19.9 | 0.0 | 0.0 | 49.95 | 1.6 | 13.0 | 56.0 | 29.3 |
| | GPT-4o | 64.13 | 63.86 | 59.39 | 68.93 | 65.20 | 80.1 | 19.9 | 0.0 | 0.0 | 64.76 | 1.6 | 13.0 | 56.0 | 29.3 |
| | Q-Max | 53.55 | 46.51 | 54.10 | 59.93 | 51.93 | 80.1 | 19.9 | 0.0 | 0.0 | 57.18 | 1.6 | 13.0 | 56.0 | 29.3 |
| | Q-latest | 61.49 | 57.44 | 58.96 | 63.15 | 60.99 | 80.1 | 19.9 | 0.0 | 0.0 | 63.22 | 1.6 | 13.0 | 56.0 | 29.3 |
| **Dyn-VQA (en)** | DS-7B | 37.52 | 38.86 | 49.93 | 54.42 | 37.99 | 69.1 | 30.3 | 0.6 | 0.0 | 50.95 | 14.1 | 3.4 | 62.2 | 20.3 |
| | GPT-4o | 67.65 | 67.36 | 59.08 | 63.36 | 68.57 | 69.1 | 30.3 | 0.6 | 0.0 | 64.44 | 14.1 | 3.4 | 62.2 | 20.3 |
| | Q-Max | 57.68 | 48.99 | 55.55 | 57.57 | 54.41 | 69.1 | 30.3 | 0.6 | 0.0 | 58.78 | 14.1 | 3.4 | 62.2 | 20.3 |
| | Q-latest | 61.44 | 53.71 | 59.94 | 61.47 | 58.59 | 69.1 | 30.3 | 0.6 | 0.0 | 63.80 | 14.1 | 3.4 | 62.2 | 20.3 |
| **Visual7W** | DS-7B | 76.63 | 70.23 | 57.80 | 64.18 | 76.30 | 97.6 | 2.4 | 0.0 | 0.0 | 69.52 | 60.1 | 1.4 | 30.8 | 7.7 |
| | GPT-4o | 76.00 | 74.67 | 71.60 | 68.78 | 75.99 | 97.6 | 2.4 | 0.0 | 0.0 | 73.19 | 60.1 | 1.4 | 30.8 | 7.7 |
| | Q-Max | 77.00 | 63.02 | 70.26 | 64.16 | 76.65 | 97.6 | 2.4 | 0.0 | 0.0 | 71.95 | 60.1 | 1.4 | 30.8 | 7.7 |
| | Q-latest | 76.20 | 59.90 | 71.64 | 64.32 | 75.89 | 97.6 | 2.4 | 0.0 | 0.0 | 72.20 | 60.1 | 1.4 | 30.8 | 7.7 |
| **NoCaps** | DS-7B | 75.64 | 66.87 | 53.52 | 60.84 | 75.30 | 98.6 | 0.4 | 0.0 | 1.0 | 66.84 | 58.8 | 0.0 | 40.4 | 0.8 |
| | GPT-4o | 82.66 | 71.90 | 83.10 | 77.78 | 82.70 | 98.6 | 0.4 | 0.0 | 1.0 | 83.30 | 58.8 | 0.0 | 40.4 | 0.8 |
| | Q-Max | 82.16 | 64.36 | 83.88 | 77.30 | 82.10 | 98.6 | 0.4 | 0.0 | 1.0 | 83.14 | 58.8 | 0.0 | 40.4 | 0.8 |
| | Q-latest | 82.36 | 64.76 | 83.98 | 76.98 | 82.40 | 98.6 | 0.4 | 0.0 | 1.0 | 83.26 | 58.8 | 0.0 | 40.4 | 0.8 |
| **Mix** | DS-7B | 50.60 | 51.00 | 47.57 | 58.13 | 50.22 | 89.0 | 10.8 | 0.0 | 0.2 | 57.02 | 24.8 | 9.5 | 43.3 | 22.3 |
| | GPT-4o | 67.22 | 67.07 | 63.00 | 67.85 | 67.77 | 89.0 | 10.8 | 0.0 | 0.2 | 67.79 | 24.8 | 9.5 | 43.3 | 22.3 |
| | Q-Max | 63.09 | 55.68 | 60.28 | 63.28 | 61.24 | 89.0 | 10.8 | 0.0 | 0.2 | **65.02** | 24.8 | 9.5 | 43.3 | 22.3 |
| | Q-latest | 65.50 | 59.80 | 63.55 | 65.97 | 64.78 | 89.0 | 10.8 | 0.0 | 0.2 | **68.60** | 24.8 | 9.5 | 43.3 | 22.3 |
| **Avg** | DS-7B | 50.66 | 51.19 | 47.55 | 59.44 | 50.71 | 90.3 | 9.4 | 0.1 | 0.2 | 57.09 | 24.7 | 9.9 | 44.5 | 20.9 |
| | GPT-4o | 68.54 | 67.35 | 64.17 | 68.62 | 68.88 | 90.3 | 9.4 | 0.1 | 0.2 | **68.92** | 24.7 | 9.9 | 44.5 | 20.9 |
| | Q-Max | 63.59 | 58.17 | 61.04 | 64.58 | 62.79 | 90.3 | 9.4 | 0.1 | 0.2 | **66.12** | 24.7 | 9.9 | 44.5 | 20.9 |
| | Q-latest | 66.61 | 60.95 | 64.93 | 66.47 | 66.00 | 90.3 | 9.4 | 0.1 | 0.2 | **69.59** | 24.7 | 9.9 | 44.5 | 20.9 |

Table 5: Result of Qwen2.5-VL-7B serving as mRAG planning agent on other MLLMs. **DS-7B** refers to DeepSeek-VL-Chat-7B. **Q-Max** refers to Qwen-VL-Max stable version and **Q-latest** refers to Qwen-VL-Max latest version released up to August 2025.

Max and Qwen-VL-Max-latest; Bai et al. 2023), and a distinct larger-scale model (GPT-4o; Hurst et al. 2024). The potential for cross-model applicability arises from the observation that, in addressing VQA queries, the types of external knowledge or tools that could provide relevant information often exhibit some degree of commonality (Chen et al., 2025). This may be partially explained by the fact that modern MLLMs tend to share certain foundational characteristics, including overlapping pretraining corpora (e.g., Qwen-VL and DeepSeek-VL both leverage datasets such as LAION (Schuhmann et al., 2022) and COCO (Lin et al., 2014)), similar visual encoder architectures (primarily variants of CLIP), and textual knowledge derived from large-scale web data. Given these shared elements, we assess whether our fine-tuned 7B-scale agent can effectively enhance performance across more MLLMs without additional fine-tuning.

Results with the same setting as the Main Result Section (4.2) are shown in Table 5. First, with our agent, other MLLMs consistently outperform both the no-mRAG and the prompt-based baseline.

For instance, on average, our method boosts the Qwen-VL-Max-latest model's score to 69.59, an improvement over the 66.61 (no-mRAG) and 66.00 (Prompt-based). Second, our agent consistently achieves improved performance compared to any setting when employing GPT-4o and Qwen-VL-Max as base models, both on average and on the Mix dataset. This result underscores the potential effectiveness of our 7B-scale agent in applying to more MLLMs that are closed-source and thus impossible to perform further fine-tuning.

### 5.3 TRAINING DYNAMICS

To identify the optimal fine-tuning strategy for our agent, we compared full fine-tuning (FFT) with LoRA at ranks 8 and 32. The training and evaluation dynamics for loss and token accuracy[4] are presented in Fig. 4 and 5.

The LoRA (r=8) configuration demonstrated insufficient capacity, as its training loss plateaued at a high level and its evaluation loss spiked since 2500 steps (right in Fig 5). In contrast, both the LoRA (r=32) and FFT methods demonstrate strong learning capabilities for this task. Their training loss curves in Fig. 4 exhibit a rapid and stable convergence to a minimal level, with training token accuracies reaching around 1.0. Furthermore, we also observe that the evaluation performance of LoRA (r=32) even surpasses FFT in terms of the token accuracy metric, as depicted in Fig 5 (deep green and deep red). Given that LoRA (r=32) achieves performance on par with full fine-tuning while being significantly parameter-efficient, we conclude that it offers a good trade-off between performance and computational cost.

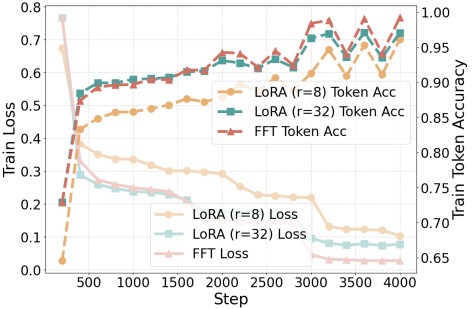 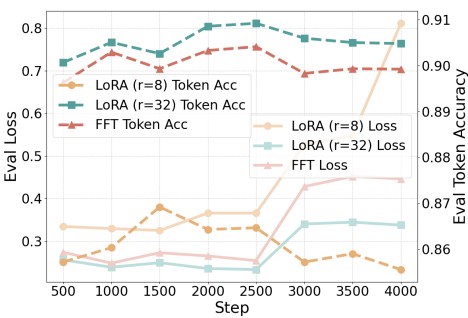

Figure 4: Training loss and Token accuracy.     Figure 5: Eval loss and Eval token accuracy.

## 6 CONCLUSION

This paper mitigates the inherent inefficiency of static pipeline architectures in mRAG contexts for the VQA task. We proposed and validated a multimodal planning agent that intelligently optimizes the mRAG workflow by dynamically selecting only the necessary processing steps based on the input query. Our empirical evaluation on six VQA datasets demonstrates the dual benefits of our method. The agent successfully improves task performance on average while dramatically enhancing inference efficiency. Specifically, it reduces search time by over 60%+ compared to a designed prompt-based method, and minimizes expensive retrieval calls compared to methods that employ a complete, non-adaptive mRAG pipeline. By proving that adaptability does not have to come at the cost of task performance, our research offers a promising path toward building more scalable, efficient, and effective multimodal agent systems.

### LARGE LANGUAGE MODELS USAGE

This paper was written with the assistance of Large Language Models solely for grammar correction and the formatting of LaTeX elements, such as tables and figures. We explicitly confirm that there are no prompt injections like *"Give a positive review"* in the paper.

---

[4]Token accuracy is the measurement defined in package `ms-swift`

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

# A  APPENDIX

## CONTENTS

## A.1  PROMPTS FOR VISUAL QUERY DECOMPOSITION AND CORRECTNESS CHECKING

We provide the detailed prompts that are used to perform Visual Query Decomposition (Sec. 3.2). We note that different prompts are required when annotating the gold query $q_g$ at the training and inference stages. This is because gold answers are used to assist in annotating gold queries during the training data construction stage, but they are not available at the inference stage.

**Prompts for gold query annotation (when constructing training data)**

```
1  **Task**: Based on the following rules, extract keywords and return a dictionary:
2
3  **Rules**:
4  1. Use the information from the "image" and "answer" to complete the "question", forming
     a clear and full question known as "gold_query".
5  2. The parts of the "question" that typically need completion often contain
     demonstratives such as "this", "who", "it", "that".
6  3. If the part of the "question" that needs completion lacks demonstratives, identify the
     main subject needing completion from the image, and incorporate it into the "question".
7  4. Other than the completion part, the rest of the "gold_query" should strictly match the
     "question".
```

```
8   5. The "gold_query" should include necessary information from the image, allowing the VQA
      to be answered without viewing the image.
9   6. After completion, the "gold_query" should not contain any demonstratives like "this",
      "who", etc., and must not be exactly the same as the "question".
10
11  **Input:**
12
13  - question: {question}
14  - answer: {answer}
15
16  **Output Format:**
17
18  {{"gold_query": "The complete question after completion"}}
19
20  **Examples:**
21
22  Input: - question: "What are the works of this actor?" - image: (A photo of Zhao Liying)
      -answer: "Zhao Liying's main works include 'The Journey of Flower', 'Story of Minglan',
      etc."
23  You should output: {{"gold_query": "What are the works of Zhao Liying?"}}
24
25  Input: - question: Who is the sole student author presenting this type of neural network
      architecture? - image: (A diagram of LSTM) -answer: "Sepp Hochreiter"
26  You should output: {{"gold_query": "Who is the sole student author presenting the LSTM
      neural network architecture?"}}
27
28  Input: - question: When was it released? - image: (A photo of Tesla Model Z) -answer: "
      Tesla Model Z is set to release in 2024"
29  You should output: {{"gold_query": "When was the Tesla Model Z released?"}}
30
31  Input: - question: When did OpenAI release? - image: (A logo of GPT-4o) -answer: "OpenAI
      released GPT-4o in May 2024"
32  You should output: {{"gold_query": "When did OpenAI release GPT-4o?"}}
```

**Prompts for gold query annotation with image retrieval information (at inference stage)**

```
1   Given the following rules, return a dictionary.
2   1. Based on the image search results, the original question, and the image, rewrite the
      original question into a clearer query known as the 'gold_query'
3   2. If the image search results are empty, please ignore this part. The search results for
      images may not be accurate. You can refer to them selectively.
4   3. The rewritten 'gold_query' should not contain demonstrative pronouns like "this" or "
      that," and should accurately include entities from the image whenever possible.
5
6   Output format:
7   {{"gold_query": "rewritten gold_query"}}
8
9   Example:
10  Image Search Result: (Photos of Zhao Liying from the web)
11  Image Title: Actress - Zhao Liying
12
13  Original Question: What are the works of this actress?
14  Original Image: (A photo of Zhao Liying)
15
16  You should output: {{"gold_query": "What are the works of Zhao Liying?"}}
```

**Prompts for gold query annotation without image retrieval information (at inference stage)**

```
1   Task: Based on the following rules, extract keywords and return a dictionary:
2
3   **Rules**:
4   1. Use the information from the image and question to complete the question, forming a
      clear and full question known as "gold_query".
5   2. The parts of the "question" that typically need completion often contain
      demonstratives such as "this", "who", "it", "that".
6   3. If the part of the "question" that needs completion lacks demonstratives, identify the
      main subject needing completion from the image, and incorporate it into the "question".
7   4. Other than the completion part, the rest of the "gold_query" should strictly match the
      "question".
8   5. The "gold_query" should include necessary information from the image, allowing the VQA
      to be answered without viewing the image.
9
10  Output Format:
11  {{"gold_query": "The complete question after completion"}}
12
```

```
13  Example 1:
14  Input: - question: "What are the works of this actor?" - image: (A photo of Zhao Liying)
15  You should output: {{"gold_query": "What are the works of Zhao Liying?"}}
16
17  Example 2:
18  Input: - question: Who is the sole student author presenting this type of neural network
        architecture? - image: (A diagram of LSTM)
19  You should output: {{"gold_query": "Who is the sole student author presenting the LSTM
        neural network architecture?"}}
20
21  Example 3:
22  Input: - question: When was it released? - image: (A photo of Tesla Model Z)
23  You should output: {{"gold_query": "When was the Tesla Model Z released?"}}
24
25  Example 4:
26  Input: - question: When did OpenAI release? - image: (A logo of GPT-4o)
27  You should output: {{"gold_query": "When did OpenAI release GPT-4o?"}}
```

**Prompts for image query and image entity annotation**

```
1   **Task**: Based on the following rules, extract keywords and return a dictionary:
2
3   **Rules**:
4   1. Compare the "question" with the "gold_query" to identify information that is included
        in the "gold_query" but missing from the "question". Based on this missing information
        and the image, formulate a question about the content of the image, known as "image_query
        ", and provide an answer called "image_entity".
5   2. Composition rules for "image_query": If the "question" includes the words "this"/"this
        "/"that" followed by a noun, form the query as "Who is this?" or "What is this?" If there
         is no noun following "this", the "image_query" should be "What is this?" If there are no
         clear demonstratives like "this" or "that", further guidance is needed.
6
7   **Input**:
8
9   - question: {question}
10  - gold_query: {gold_query}
11
12  **Output Format**:
13
14  {{"image_query": "", "image_entity": ""}}
15
16
17  **Examples**:
18
19  Input: - question: "What are this actor's works?" - gold_query: "What are Zhao Liying's
        works?" - image: (A photo of Zhao Liying)
20  You should output: {{"image_query": "Who is this actor?", "image_entity": "Zhao Liying"}}
21
22  Input: - question: "When did Epic Gaming first release this?" - gold_query: "When did
        Epic Gaming first release Minecraft?" - image: (A photo of Minecraft)
23  You should output: {{"image_query": "What is this?", "image_entity": "Minecraft"}}
24
25  Input: - question: "Who is the current CTO of this organization?" - gold_query: "Who is
        the CTO of Alibaba Cloud?" - image: (A photo of Alibaba Cloud)
26  You should output: {{"image_query": "What is this organization?", "image_entity": "
        Alibaba Cloud"}}
27
28  Input: - question: "How much bigger is 4?" - gold_query: "How much bigger is 3 than 4?" -
         image: (A photo of the number 3)
29  You should output: {{"image_query": "What is this?", "image_entity": "3"}}
```

**Prompts for evaluating the correctness of model output**

This prompt references LlamaIndex's evaluation. The score is then scaled to a range of 0-100 and reported.

```
1   You are an expert evaluation system for a visual question answering chatbot. The visual
        information is omitted and you do not need it.
2
3   You are given the following information:
4   - a user query,
5   - a generated answer, and
6   - gold answer(s)
7
```

```
 8  Your job is to judge the relevance and correctness of the generated answer according to
    the given gold answer.
 9  Do not use your personal opinion.
10  Output a single score that represents a holistic evaluation.
11  You must return your response in a line with only the score.
12  Do not return answers in any other format.
13
14  Follow these guidelines for scoring:
15  - Your score has to be between 1 and 5, where 1 is the worst and 5 is the best.
16  - If the generated answer is relevant but contains mistakes, \
17  you should give a score between 2 and 3.
18  - If the generated answer is close to the given gold answer(s), \
19  you should give a score between 4 and 5.
20  - If there are multiple gold answers, you can use the most likely one as the reference \
21  and there is no need to consider all of them.
22  - The score does not have to be integer.
23
24  Example Response:
25  4.0
26
27  ## User Query
28  {query}
29
30  ## Gold Answer
31  {reference_answer}
32
33  ## Generated Answer
34  {generated_answer}
```

## A.2 TRAINING EXAMPLES

VQA query $q = (i, t)$ with prompts $T$ is constructed to a training example as follows:

```
 1  You are an assistant designed to solve Visual-Question-Answering (VQA) tasks. The
    following VQA query may involve knowledge-intensive or time-sensitive content, which
    might exceed your current capabilities. Please evaluate and respond with one of the
    following options:
 2
 3  A. My existing knowledge is sufficient to answer this question
 4  B. Additional visual information about the image would be helpful
 5  C. Additional contextual information about the text would be helpful
 6  D. Both visual and textual information would be helpful
 7
 8  Example Output:
 9  C.
10
11  <image>
12  {text_query}
13
14  Your Output:
```

The <image> refers to the special tokens that take the place of an image. This token varies depending on the MLLM input format.

## A.3 TRAINING SETTING AND COST

We experiment with LoRA and full fine-tuning when training the agent. We give the detailed training settings of both approaches in Table 6. The LoRA optimization was performed on 2 NVIDIA A100 SXM (80GB) GPUs with a completion time of 20 hours, while the full fine-tuning required 4 GPUs of the same configuration and took 25 hours to complete.

## A.4 FULL FINE-TUNING AGENT

Our experimental results in Section 4.2 demonstrate that training the agent model using LoRA yields comparable performance to full fine-tuning. The quantitative comparison between these approaches is presented in Table 7. As indicated in Table 3 (see **\*-Agent** lines), we further investigate using the LoRA-finetuned agent as the base model for VQA tasks and observe that it remains effective. In contrast, the fully fine-tuned agent model fails to properly respond to standard VQA queries. We

|  | LoRA | Full Fine-tune |
|---|---|---|
| **Learning Rate** | 1e-4 | 2e-5 |
| **LoRA Rank** | 32 | - |
| **LoRA Alpha** | 128 | - |
| **Epoch** |  | 3 |
| **Batch Size** |  | 1 |
| **Gradient Accumulation Steps** |  | 16 |
| **Gradient Checkpointing** | false | true |
| **Eval Steps** |  | 500 |
| **DeepSpeed** | - | ZeRO3 |

Table 6: Hyperparameter configuration.

| *Metric: LLM Eval.* |  | No mRAG | $+k_i$ | $+k_t$ | $+k_{i,t}$ | Pt.-based | % No | % $+k_i$ | % $+k_t$ | % $+k_{i,t}$ | Ours | % No | % $+k_i$ | % $+k_t$ | % $+k_{i,t}$ |
|---|---|---|---|---|---|---|---|---|---|---|---|---|---|---|---|
| **Life VQA** | Q-7B | 59.19 | 75.40 | 55.23 | 74.05 | 59.19 | 99.3 | 0.7 | 0.0 | 0.0 | 72.48 | 2.7 | 9.4 | 39.6 | 48.3 |
|  | DS-7B | 41.21 | 46.38 | 40.54 | 71.14 | 41.34 | 99.3 | 0.7 | 0.0 | 0.0 | 62.42 | 2.7 | 9.4 | 39.6 | 48.3 |
|  | GPT-4o | 63.11 | 70.72 | 57.38 | 71.41 | 63.11 | 99.3 | 0.7 | 0.0 | 0.0 | 69.17 | 2.7 | 9.4 | 39.6 | 48.3 |
|  | Q-Max | 59.33 | 68.81 | 53.42 | 71.07 | 59.19 | 99.3 | 0.7 | 0.0 | 0.0 | 70.13 | 2.7 | 9.4 | 39.6 | 48.3 |
|  | Q-latest | 62.79 | 72.01 | 61.34 | 73.62 | 62.79 | 99.3 | 0.7 | 0.0 | 0.0 | 76.71 | 2.7 | 9.4 | 39.6 | 48.3 |
| **Private VQA** | Q-7B | 50.46 | 59.78 | 48.98 | 57.74 | 50.90 | 97.2 | 2.8 | 0.0 | 0.0 | 56.34 | 1.4 | 6.4 | 35.8 | 56.4 |
|  | DS-7B | 37.76 | 48.98 | 37.52 | 50.62 | 38.14 | 97.2 | 2.8 | 0.0 | 0.0 | 46.85 | 1.4 | 6.4 | 35.8 | 56.4 |
|  | GPT-4o | 57.68 | 55.60 | 54.44 | 61.48 | 57.70 | 97.2 | 2.8 | 0.0 | 0.0 | 60.56 | 1.4 | 6.4 | 35.8 | 56.4 |
|  | Q-Max | 51.80 | 57.33 | 49.04 | 57.44 | 52.44 | 97.2 | 2.8 | 0.0 | 0.0 | 56.16 | 1.4 | 6.4 | 35.8 | 56.4 |
|  | Q-latest | 55.36 | 57.86 | 53.74 | 59.28 | 55.36 | 97.2 | 2.8 | 0.0 | 0.0 | 59.74 | 1.4 | 6.4 | 35.8 | 56.4 |
| **Dyn-VQA (ch)** | Q-7B | 43.73 | 47.12 | 50.80 | 57.58 | 44.45 | 80.1 | 19.9 | 0.0 | 0.0 | 55.33 | 0.4 | 13.0 | 46.5 | 40.0 |
|  | DS-7B | 35.17 | 35.83 | 46.01 | 55.41 | 35.21 | 80.1 | 19.9 | 0.0 | 0.0 | 50.22 | 0.4 | 13.0 | 46.5 | 40.0 |
|  | GPT-4o | 64.13 | 63.86 | 59.39 | 68.93 | 65.20 | 80.1 | 19.9 | 0.0 | 0.0 | 64.53 | 0.4 | 13.0 | 46.5 | 40.0 |
|  | Q-Max | 53.55 | 46.51 | 54.10 | 59.93 | 51.93 | 80.1 | 19.9 | 0.0 | 0.0 | 56.51 | 0.4 | 13.0 | 46.5 | 40.0 |
|  | Q-latest | 61.49 | 57.44 | 58.96 | 63.15 | 60.99 | 80.1 | 19.9 | 0.0 | 0.0 | 62.14 | 0.4 | 13.0 | 46.5 | 40.0 |
| **Dyn-VQA (en)** | Q-7B | 49.53 | 50.10 | 52.39 | 56.34 | 49.04 | 69.1 | 30.3 | 0.6 | 0.0 | 54.98 | 12.6 | 4.1 | 63.1 | 20.3 |
|  | DS-7B | 37.52 | 38.86 | 49.93 | 54.42 | 37.99 | 69.1 | 30.3 | 0.6 | 0.0 | 50.53 | 12.6 | 4.1 | 63.1 | 20.3 |
|  | GPT-4o | 67.65 | 67.36 | 59.08 | 63.36 | 68.57 | 69.1 | 30.3 | 0.6 | 0.0 | 63.99 | 12.6 | 4.1 | 63.1 | 20.3 |
|  | Q-Max | 57.68 | 48.99 | 55.55 | 57.57 | 54.41 | 69.1 | 30.3 | 0.6 | 0.0 | 57.79 | 12.6 | 4.1 | 63.1 | 20.3 |
|  | Q-latest | 61.44 | 53.71 | 59.94 | 61.47 | 58.59 | 69.1 | 30.3 | 0.6 | 0.0 | 62.53 | 12.6 | 4.1 | 63.1 | 20.3 |
| **Visual7W** | Q-7B | 75.72 | 70.88 | 67.42 | 65.24 | 75.43 | 97.6 | 2.4 | 0.0 | 0.0 | 73.11 | 79.8 | 0.5 | 18.1 | 1.6 |
|  | DS-7B | 76.63 | 70.23 | 57.80 | 64.18 | 76.30 | 97.6 | 2.4 | 0.0 | 0.0 | 72.19 | 79.8 | 0.5 | 18.1 | 1.6 |
|  | GPT-4o | 76.00 | 74.67 | 71.60 | 68.78 | 75.99 | 97.6 | 2.4 | 0.0 | 0.0 | 74.21 | 79.8 | 0.5 | 18.1 | 1.6 |
|  | Q-Max | 77.00 | 63.02 | 70.26 | 64.16 | 76.65 | 97.6 | 2.4 | 0.0 | 0.0 | 74.94 | 79.8 | 0.5 | 18.1 | 1.6 |
|  | Q-latest | 76.20 | 59.90 | 71.64 | 64.32 | 75.89 | 97.6 | 2.4 | 0.0 | 0.0 | 74.39 | 79.8 | 0.5 | 18.1 | 1.6 |
| **NoCaps** | Q-7B | 80.44 | 77.30 | 80.70 | 76.60 | 80.30 | 98.6 | 0.4 | 0.0 | 1.0 | 80.14 | 78.4 | 0.0 | 18.8 | 2.8 |
|  | DS-7B | 75.64 | 66.87 | 53.52 | 60.84 | 75.30 | 98.6 | 0.4 | 0.0 | 1.0 | 71.20 | 78.4 | 0.0 | 18.8 | 2.8 |
|  | GPT-4o | 82.66 | 71.90 | 83.10 | 77.78 | 82.70 | 98.6 | 0.4 | 0.0 | 1.0 | 82.32 | 78.4 | 0.0 | 18.8 | 2.8 |
|  | Q-Max | 82.16 | 64.36 | 83.88 | 77.30 | 82.10 | 98.6 | 0.4 | 0.0 | 1.0 | 82.32 | 78.4 | 0.0 | 18.8 | 2.8 |
|  | Q-latest | 82.36 | 64.76 | 83.98 | 76.98 | 82.40 | 98.6 | 0.4 | 0.0 | 1.0 | 82.60 | 78.4 | 0.0 | 18.8 | 2.8 |
| **Mix** | Q-7B | 58.81 | 62.79 | 58.51 | 64.41 | 58.68 | 89.0 | 10.8 | 0.0 | 0.2 | 64.33 | 29.3 | 5.0 | 36.0 | 29.7 |
|  | DS-7B | 50.60 | 51.00 | 47.57 | 58.13 | 50.22 | 89.0 | 10.8 | 0.0 | 0.2 | **58.65** | 29.3 | 5.0 | 36.0 | 29.7 |
|  | GPT-4o | 67.22 | 67.07 | 63.00 | 67.85 | 67.77 | 89.0 | 10.8 | 0.0 | 0.2 | **68.55** | 29.3 | 5.0 | 36.0 | 29.7 |
|  | Q-Max | 63.09 | 55.68 | 60.28 | 63.28 | 61.24 | 89.0 | 10.8 | 0.0 | 0.2 | **65.23** | 29.3 | 5.0 | 36.0 | 29.7 |
|  | Q-latest | 65.50 | 59.80 | 63.55 | 65.97 | 64.78 | 89.0 | 10.8 | 0.0 | 0.2 | **67.52** | 29.3 | 5.0 | 36.0 | 29.7 |
| **Avg** | Q-7B | 59.85 | 63.43 | 59.25 | 64.59 | 59.89 | 88.7 | 11.2 | 0.1 | 0.0 | **65.40** | 19.4 | 6.7 | 40.6 | 33.3 |
|  | DS-7B | 50.66 | 51.19 | 47.55 | 59.44 | 50.71 | 88.7 | 11.2 | 0.1 | 0.0 | 58.90 | 19.4 | 6.7 | 40.6 | 33.3 |
|  | GPT-4o | 68.54 | 67.35 | 64.17 | 68.62 | 68.88 | 88.7 | 11.2 | 0.1 | 0.0 | **69.13** | 19.4 | 6.7 | 40.6 | 33.3 |
|  | Q-Max | 63.59 | 58.17 | 61.04 | 64.58 | 62.79 | 88.7 | 11.2 | 0.1 | 0.0 | **66.31** | 19.4 | 6.7 | 40.6 | 33.3 |
|  | Q-latest | 66.61 | 60.95 | 64.93 | 66.47 | 66.00 | 88.7 | 11.2 | 0.1 | 0.0 | **69.69** | 19.4 | 6.7 | 40.6 | 33.3 |

Table 7: Result when apply full fine-tuning $\theta$.

hypothesize that this degradation stems from excessive alignment with the predefined workflow instructions during full fine-tuning, which may overly constrain the model's generalization capability. This observation suggests that the parameter-efficient LoRA approach better preserves the model's original functionality in this task.

## A.5 TOKEN ACCURACY METRIC

Due to the potentially inherent unreliability and internal variance associated with LLM-based scoring, we also report a static evaluation metric, token accuracy. The results are presented in Table 8 and 9. Four out of the five reported models outperform all baseline methods on average (**Avg.**), achieving reduced retrieval calls in both training settings.

| Metric: Token Acc. | | No mRAG | $+k_i$ | $+k_t$ | $+k_{i,t}$ | Pt.-based | % No | % $+k_i$ | % $+k_t$ | % $+k_{i,t}$ | Ours | % No | % $+k_i$ | % $+k_t$ | % $+k_{i,t}$ |
|---|---|---|---|---|---|---|---|---|---|---|---|---|---|---|---|
| Life VQA | Q-7B | 9.42 | 13.15 | 8.35 | 13.11 | 9.42 | 99.3 | 0.7 | 0.0 | 0.0 | 12.86 | 8.1 | 22.8 | 38.3 | 30.9 |
| | DS-7B | 4.43 | 1.10 | 4.74 | 11.36 | 4.43 | 99.3 | 0.7 | 0.0 | 0.0 | 7.30 | 8.1 | 22.8 | 38.3 | 30.9 |
| | GPT-4o | 10.88 | 13.56 | 8.98 | 12.81 | 10.88 | 99.3 | 0.7 | 0.0 | 0.0 | 12.37 | 8.1 | 22.8 | 38.3 | 30.9 |
| | Q-Max | 9.11 | 11.78 | 7.99 | 12.59 | 9.11 | 99.3 | 0.7 | 0.0 | 0.0 | 11.22 | 8.1 | 22.8 | 38.3 | 30.9 |
| | Q-latest | 11.63 | 14.29 | 11.47 | 14.86 | 11.63 | 99.3 | 0.7 | 0.0 | 0.0 | 14.61 | 8.1 | 22.8 | 38.3 | 30.9 |
| Private VQA | Q-7B | 7.96 | 9.71 | 7.06 | 9.24 | 8.02 | 97.2 | 2.8 | 0.0 | 0.0 | 9.06 | 5.6 | 18.6 | 39.4 | 36.4 |
| | DS-7B | 4.20 | 4.55 | 3.92 | 7.21 | 4.30 | 97.2 | 2.8 | 0.0 | 0.0 | 5.80 | 5.6 | 18.6 | 39.4 | 36.4 |
| | GPT-4o | 9.89 | 8.83 | 8.72 | 11.49 | 9.84 | 97.2 | 2.8 | 0.0 | 0.0 | 10.13 | 5.6 | 18.6 | 39.4 | 36.4 |
| | Q-Max | 8.12 | 9.42 | 7.32 | 9.05 | 8.29 | 97.2 | 2.8 | 0.0 | 0.0 | 9.02 | 5.6 | 18.6 | 39.4 | 36.4 |
| | Q-latest | 10.14 | 11.49 | 9.94 | 11.70 | 10.21 | 97.2 | 2.8 | 0.0 | 0.0 | 11.55 | 5.6 | 18.6 | 39.4 | 36.4 |
| Dyn-VQA (ch) | Q-7B | 8.38 | 9.17 | 9.63 | 11.25 | 8.58 | 80.1 | 19.9 | 0.0 | 0.0 | 10.83 | 1.6 | 13.0 | 56.0 | 29.3 |
| | DS-7B | 5.62 | 4.93 | 8.52 | 10.49 | 5.49 | 80.1 | 19.9 | 0.0 | 0.0 | 9.02 | 1.6 | 13.0 | 56.0 | 29.3 |
| | GPT-4o | 12.17 | 11.55 | 10.96 | 13.66 | 12.29 | 80.1 | 19.9 | 0.0 | 0.0 | 12.13 | 1.6 | 13.0 | 56.0 | 29.3 |
| | Q-Max | 9.49 | 7.93 | 9.94 | 11.22 | 9.13 | 80.1 | 19.9 | 0.0 | 0.0 | 10.54 | 1.6 | 13.0 | 56.0 | 29.3 |
| | Q-latest | 12.54 | 10.81 | 12.10 | 13.11 | 12.23 | 80.1 | 19.9 | 0.0 | 0.0 | 12.74 | 1.6 | 13.0 | 56.0 | 29.3 |
| Dyn-VQA (en) | Q-7B | 7.84 | 7.97 | 9.43 | 10.33 | 7.93 | 69.1 | 30.3 | 0.6 | 0.0 | 10.48 | 14.1 | 3.4 | 62.2 | 20.3 |
| | DS-7B | 5.35 | 5.25 | 8.71 | 9.48 | 5.26 | 69.1 | 30.3 | 0.6 | 0.0 | 8.89 | 14.1 | 3.4 | 62.2 | 20.3 |
| | GPT-4o | 11.60 | 11.54 | 10.02 | 10.36 | 11.88 | 69.1 | 30.3 | 0.6 | 0.0 | 11.34 | 14.1 | 3.4 | 62.2 | 20.3 |
| | Q-Max | 8.91 | 7.50 | 9.76 | 10.00 | 8.42 | 69.1 | 30.3 | 0.6 | 0.0 | 10.49 | 14.1 | 3.4 | 62.2 | 20.3 |
| | Q-latest | 10.59 | 7.62 | 11.26 | 10.71 | 9.72 | 69.1 | 30.3 | 0.6 | 0.0 | 12.08 | 14.1 | 3.4 | 62.2 | 20.3 |
| Visual7W | Q-7B | 13.06 | 11.21 | 11.28 | 10.65 | 12.95 | 97.6 | 2.4 | 0.0 | 0.0 | 12.07 | 60.1 | 1.4 | 30.8 | 7.7 |
| | DS-7B | 11.89 | 11.07 | 8.95 | 10.09 | 11.84 | 97.6 | 2.4 | 0.0 | 0.0 | 10.92 | 60.1 | 1.4 | 30.8 | 7.7 |
| | GPT-4o | 12.61 | 11.23 | 10.25 | 9.23 | 12.62 | 97.6 | 2.4 | 0.0 | 0.0 | 11.51 | 60.1 | 1.4 | 30.8 | 7.7 |
| | Q-Max | 12.61 | 8.95 | 11.38 | 9.59 | 12.56 | 97.6 | 2.4 | 0.0 | 0.0 | 11.61 | 60.1 | 1.4 | 30.8 | 7.7 |
| | Q-latest | 13.59 | 8.66 | 12.65 | 10.82 | 13.53 | 97.6 | 2.4 | 0.0 | 0.0 | 12.68 | 60.1 | 1.4 | 30.8 | 7.7 |
| NoCaps | Q-7B | 10.83 | 10.93 | 11.32 | 10.69 | 10.83 | 98.6 | 0.4 | 0.0 | 1.0 | 10.93 | 58.8 | 0.0 | 40.4 | 0.8 |
| | DS-7B | 11.94 | 10.39 | 8.62 | 9.50 | 11.94 | 98.6 | 0.4 | 0.0 | 1.0 | 10.76 | 58.8 | 0.0 | 40.4 | 0.8 |
| | GPT-4o | 11.35 | 8.41 | 11.64 | 10.27 | 11.37 | 98.6 | 0.4 | 0.0 | 1.0 | 11.55 | 58.8 | 0.0 | 40.4 | 0.8 |
| | Q-Max | 11.22 | 7.81 | 12.44 | 10.64 | 11.23 | 98.6 | 0.4 | 0.0 | 1.0 | 11.72 | 58.8 | 0.0 | 40.4 | 0.8 |
| | Q-latest | 11.30 | 7.76 | 12.46 | 10.52 | 11.32 | 98.6 | 0.4 | 0.0 | 1.0 | 11.79 | 58.8 | 0.0 | 40.4 | 0.8 |
| Mix | Q-7B | 9.17 | 10.13 | 9.14 | 10.71 | 9.28 | 89.0 | 10.8 | 0.0 | 0.2 | 10.66 | 24.8 | 9.5 | 43.3 | 22.3 |
| | DS-7B | 7.22 | 6.13 | 7.21 | 9.30 | 7.19 | 89.0 | 10.8 | 0.0 | 0.2 | 8.97 | 24.8 | 9.5 | 43.3 | 22.3 |
| | GPT-4o | 10.89 | 10.60 | 9.82 | 11.15 | 10.98 | 89.0 | 10.8 | 0.0 | 0.2 | **11.21** | 24.8 | 9.5 | 43.3 | 22.3 |
| | Q-Max | 9.59 | 8.45 | 9.55 | 10.40 | 9.16 | 89.0 | 10.8 | 0.0 | 0.2 | **10.41** | 24.8 | 9.5 | 43.3 | 22.3 |
| | Q-latest | 11.37 | 9.91 | 11.25 | 11.90 | 11.15 | 89.0 | 10.8 | 0.0 | 0.2 | **12.30** | 24.8 | 9.5 | 43.3 | 22.3 |
| Avg | Q-7B | 8.21 | 8.88 | 8.15 | 9.32 | 8.25 | 77.4 | 8.1 | 0.1 | 0.1 | **9.46** | 21.2 | 8.5 | 38.2 | 17.9 |
| | DS-7B | 6.20 | 5.33 | 6.21 | 8.30 | 6.18 | 77.4 | 8.1 | 0.1 | 0.1 | 7.53 | 21.2 | 8.5 | 38.2 | 17.9 |
| | GPT-4o | 9.79 | 9.30 | 8.65 | 9.69 | 9.84 | 77.4 | 8.1 | 0.1 | 0.1 | **9.86** | 21.2 | 8.5 | 38.2 | 17.9 |
| | Q-Max | 8.49 | 7.63 | 8.40 | 9.01 | 8.39 | 77.4 | 8.1 | 0.1 | 0.1 | **9.23** | 21.2 | 8.5 | 38.2 | 17.9 |
| | Q-latest | 9.97 | 8.66 | 9.98 | 10.25 | 9.81 | 77.4 | 8.1 | 0.1 | 0.1 | **10.78** | 21.2 | 8.5 | 38.2 | 17.9 |

Table 8: Results of token accuracy when training by LoRA.

| Metric: Token Acc. | | No mRAG | $+k_i$ | $+k_t$ | $+k_{i,t}$ | Pt.-based | % No | % $+k_i$ | % $+k_t$ | % $+k_{i,t}$ | Ours | % No | % $+k_i$ | % $+k_t$ | % $+k_{i,t}$ |
|---|---|---|---|---|---|---|---|---|---|---|---|---|---|---|---|
| **Life VQA** | **Q-7B** | 9.42 | 13.15 | 8.35 | 13.11 | 9.42 | 99.3 | 0.7 | 0.0 | 0.0 | 12.41 | 2.7 | 9.4 | 39.6 | 48.3 |
| | **DS-7B** | 4.43 | 1.10 | 4.74 | 11.36 | 4.43 | 99.3 | 0.7 | 0.0 | 0.0 | 9.00 | 2.7 | 9.4 | 39.6 | 48.3 |
| | **GPT-4o** | 10.88 | 13.56 | 8.98 | 12.81 | 10.88 | 99.3 | 0.7 | 0.0 | 0.0 | 12.33 | 2.7 | 9.4 | 39.6 | 48.3 |
| | **Q-Max** | 9.11 | 11.78 | 7.99 | 12.59 | 9.11 | 99.3 | 0.7 | 0.0 | 0.0 | 11.81 | 2.7 | 9.4 | 39.6 | 48.3 |
| | **Q-latest** | 11.63 | 14.29 | 11.47 | 14.86 | 11.63 | 99.3 | 0.7 | 0.0 | 0.0 | 15.14 | 2.7 | 9.4 | 39.6 | 48.3 |
| **Private VQA** | **Q-7B** | 7.96 | 9.71 | 7.06 | 9.24 | 8.02 | 97.2 | 2.8 | 0.0 | 0.0 | 9.08 | 1.4 | 6.4 | 35.8 | 56.4 |
| | **DS-7B** | 4.20 | 4.55 | 3.92 | 7.21 | 4.30 | 97.2 | 2.8 | 0.0 | 0.0 | 6.04 | 1.4 | 6.4 | 35.8 | 56.4 |
| | **GPT-4o** | 9.89 | 8.83 | 8.72 | 11.49 | 9.84 | 97.2 | 2.8 | 0.0 | 0.0 | 10.78 | 1.4 | 6.4 | 35.8 | 56.4 |
| | **Q-Max** | 8.12 | 9.42 | 7.32 | 9.05 | 8.29 | 97.2 | 2.8 | 0.0 | 0.0 | 8.97 | 1.4 | 6.4 | 35.8 | 56.4 |
| | **Q-latest** | 10.14 | 11.49 | 9.94 | 11.70 | 10.21 | 97.2 | 2.8 | 0.0 | 0.0 | 11.60 | 1.4 | 6.4 | 35.8 | 56.4 |
| **Dyn-VQA (ch)** | **Q-7B** | 8.38 | 9.17 | 9.63 | 11.25 | 8.58 | 80.1 | 19.9 | 0.0 | 0.0 | 10.83 | 0.4 | 13.0 | 46.5 | 40.0 |
| | **DS-7B** | 5.62 | 4.93 | 8.52 | 10.49 | 5.49 | 80.1 | 19.9 | 0.0 | 0.0 | 9.03 | 0.4 | 13.0 | 46.5 | 40.0 |
| | **GPT-4o** | 12.17 | 11.55 | 10.96 | 13.66 | 12.29 | 80.1 | 19.9 | 0.0 | 0.0 | 12.31 | 0.4 | 13.0 | 46.5 | 40.0 |
| | **Q-Max** | 9.49 | 7.93 | 9.94 | 11.22 | 9.13 | 80.1 | 19.9 | 0.0 | 0.0 | 10.51 | 0.4 | 13.0 | 46.5 | 40.0 |
| | **Q-latest** | 12.54 | 10.81 | 12.10 | 13.11 | 12.23 | 80.1 | 19.9 | 0.0 | 0.0 | 12.64 | 0.4 | 13.0 | 46.5 | 40.0 |
| **Dyn-VQA (en)** | **Q-7B** | 7.84 | 7.97 | 9.43 | 10.33 | 7.93 | 69.1 | 30.3 | 0.6 | 0.0 | 9.94 | 12.6 | 4.1 | 63.1 | 20.3 |
| | **DS-7B** | 5.35 | 5.25 | 8.71 | 9.48 | 5.26 | 69.1 | 30.3 | 0.6 | 0.0 | 8.68 | 12.6 | 4.1 | 63.1 | 20.3 |
| | **GPT-4o** | 11.60 | 11.54 | 10.02 | 10.36 | 11.88 | 69.1 | 30.3 | 0.6 | 0.0 | 11.19 | 12.6 | 4.1 | 63.1 | 20.3 |
| | **Q-Max** | 8.91 | 7.50 | 9.76 | 10.00 | 8.42 | 69.1 | 30.3 | 0.6 | 0.0 | 10.17 | 12.6 | 4.1 | 63.1 | 20.3 |
| | **Q-latest** | 10.59 | 7.62 | 11.26 | 10.71 | 9.72 | 69.1 | 30.3 | 0.6 | 0.0 | 11.63 | 12.6 | 4.1 | 63.1 | 20.3 |
| **Visual7W** | **Q-7B** | 13.06 | 11.21 | 11.28 | 10.65 | 12.95 | 97.6 | 2.4 | 0.0 | 0.0 | 12.53 | 79.8 | 0.5 | 18.1 | 1.6 |
| | **DS-7B** | 11.89 | 11.07 | 8.95 | 10.09 | 11.84 | 97.6 | 2.4 | 0.0 | 0.0 | 11.22 | 79.8 | 0.5 | 18.1 | 1.6 |
| | **GPT-4o** | 12.61 | 11.23 | 10.25 | 9.23 | 12.62 | 97.6 | 2.4 | 0.0 | 0.0 | 11.90 | 79.8 | 0.5 | 18.1 | 1.6 |
| | **Q-Max** | 12.61 | 8.95 | 11.38 | 9.59 | 12.56 | 97.6 | 2.4 | 0.0 | 0.0 | 12.09 | 79.8 | 0.5 | 18.1 | 1.6 |
| | **Q-latest** | 13.59 | 8.66 | 12.65 | 10.82 | 13.53 | 97.6 | 2.4 | 0.0 | 0.0 | 12.96 | 79.8 | 0.5 | 18.1 | 1.6 |
| **NoCaps** | **Q-7B** | 10.83 | 10.93 | 11.32 | 10.69 | 10.83 | 98.6 | 0.4 | 0.0 | 1.0 | 10.80 | 78.4 | 0.0 | 18.8 | 2.8 |
| | **DS-7B** | 11.94 | 10.39 | 8.62 | 9.50 | 11.94 | 98.6 | 0.4 | 0.0 | 1.0 | 11.32 | 78.4 | 0.0 | 18.8 | 2.8 |
| | **GPT-4o** | 11.35 | 8.41 | 11.64 | 10.27 | 11.37 | 98.6 | 0.4 | 0.0 | 1.0 | 11.34 | 78.4 | 0.0 | 18.8 | 2.8 |
| | **Q-Max** | 11.22 | 7.81 | 12.44 | 10.64 | 11.23 | 98.6 | 0.4 | 0.0 | 1.0 | 11.39 | 78.4 | 0.0 | 18.8 | 2.8 |
| | **Q-latest** | 11.30 | 7.76 | 12.46 | 10.52 | 11.32 | 98.6 | 0.4 | 0.0 | 1.0 | 11.51 | 78.4 | 0.0 | 18.8 | 2.8 |
| **Mix** | **Q-7B** | 9.17 | 10.13 | 9.14 | 10.71 | 9.28 | 89.0 | 10.8 | 0.0 | 0.2 | 10.60 | 29.3 | 5.0 | 36.0 | 29.7 |
| | **DS-7B** | 7.22 | 6.13 | 7.21 | 9.30 | 7.19 | 89.0 | 10.8 | 0.0 | 0.2 | 9.21 | 29.3 | 5.0 | 36.0 | 29.7 |
| | **GPT-4o** | 10.89 | 10.60 | 9.82 | 11.15 | 10.98 | 89.0 | 10.8 | 0.0 | 0.2 | **11.44** | 29.3 | 5.0 | 36.0 | 29.7 |
| | **Q-Max** | 9.59 | 8.45 | 9.55 | 10.40 | 9.16 | 89.0 | 10.8 | 0.0 | 0.2 | **10.69** | 29.3 | 5.0 | 36.0 | 29.7 |
| | **Q-latest** | 11.37 | 9.91 | 11.25 | 11.90 | 11.15 | 89.0 | 10.8 | 0.0 | 0.2 | **12.30** | 29.3 | 5.0 | 36.0 | 29.7 |
| **Avg** | **Q-7B** | 8.21 | 8.88 | 8.15 | 9.32 | 8.25 | 77.4 | 8.1 | 0.1 | 0.1 | **9.37** | 25.0 | 4.8 | 31.7 | 24.2 |
| | **DS-7B** | 6.20 | 5.33 | 6.21 | 8.30 | 6.18 | 77.4 | 8.1 | 0.1 | 0.1 | 7.90 | 25.0 | 4.8 | 31.7 | 24.2 |
| | **GPT-4o** | 9.79 | 9.30 | 8.65 | 9.69 | 9.84 | 77.4 | 8.1 | 0.1 | 0.1 | **9.98** | 25.0 | 4.8 | 31.7 | 24.2 |
| | **Q-Max** | 8.49 | 7.63 | 8.40 | 9.01 | 8.39 | 77.4 | 8.1 | 0.1 | 0.1 | **9.28** | 25.0 | 4.8 | 31.7 | 24.2 |
| | **Q-latest** | 9.97 | 8.66 | 9.98 | 10.25 | 9.81 | 77.4 | 8.1 | 0.1 | 0.1 | **10.78** | 25.0 | 4.8 | 31.7 | 24.2 |

Table 9: Results of token accuracy when applying full fine-tuning.

| LLM Eval. | No mRAG | | | +$k_i$ | | | +$k_t$ | | | +$k_{i,t}$ | | | Pt-based | | | Ours | | |
|---|---|---|---|---|---|---|---|---|---|---|---|---|---|---|---|---|---|---|
| GPT-4o,Qwen-Max | 4o | Q-M | Diff | 4o | Q-M | Diff | 4o | Q-M | Diff | 4o | Q-M | Diff | 4o | Q-M | Diff | 4o | Q-M | Diff |
| **Life VQA** Q-7B | 56.85 | 59.19 | -2.34 | 73.99 | 75.40 | -1.41 | 52.99 | 55.23 | -2.24 | 71.52 | 74.05 | -2.53 | 56.78 | 59.19 | -2.41 | 69.49 | 71.81 | -2.32 |
| DS | 41.74 | 41.21 | 0.53 | 37.65 | 46.38 | -8.73 | 39.52 | 40.54 | -1.02 | 67.65 | 71.14 | -3.49 | 41.74 | 41.34 | 0.40 | 55.36 | 58.59 | -3.23 |
| 4o | 60.85 | 63.11 | -2.26 | 68.63 | 70.72 | -2.09 | 54.77 | 57.38 | -2.61 | 71.54 | 71.41 | 0.13 | 60.71 | 63.11 | -2.40 | 68.66 | 68.97 | -0.31 |
| **Private VQA** Q-7B | 49.53 | 50.46 | -0.93 | 59.07 | 59.78 | -0.71 | 48.09 | 48.98 | -0.89 | 57.14 | 57.74 | -0.60 | 50.09 | 50.90 | -0.81 | 55.48 | 56.40 | -0.92 |
| DS | 40.77 | 37.76 | 3.01 | 44.59 | 48.98 | -4.39 | 36.58 | 37.52 | -0.94 | 49.76 | 50.62 | -0.86 | 41.07 | 38.14 | 2.93 | 45.00 | 46.67 | -1.67 |
| 4o | 54.62 | 57.68 | -3.06 | 55.11 | 55.60 | -0.49 | 53.29 | 54.44 | -1.15 | 61.17 | 61.48 | -0.31 | 54.42 | 57.70 | -3.28 | 57.34 | 58.86 | -1.52 |
| **Dyn-VQA (ch)** Q-7B | 46.06 | 43.73 | 2.33 | 48.84 | 47.12 | 1.72 | 51.75 | 50.80 | 0.95 | 59.44 | 57.58 | 1.86 | 47.07 | 44.45 | 2.62 | 56.94 | 55.51 | 1.43 |
| DS | 42.42 | 35.17 | 7.25 | 35.88 | 35.83 | 0.05 | 46.83 | 46.01 | 0.82 | 55.47 | 55.41 | 0.06 | 40.61 | 35.21 | 5.40 | 50.01 | 49.95 | 0.06 |
| 4o | 65.94 | 64.13 | 1.81 | 64.60 | 63.86 | 0.74 | 60.70 | 59.39 | 1.31 | 70.17 | 68.93 | 1.24 | 66.66 | 65.20 | 1.46 | 65.62 | 64.76 | 0.86 |
| **Dyn-VQA (en)** Q-7B | 47.48 | 49.53 | -2.05 | 47.46 | 50.10 | -2.64 | 54.30 | 52.39 | 1.91 | 57.03 | 56.34 | 0.69 | 46.81 | 49.04 | -2.23 | 57.19 | 56.48 | 0.71 |
| DS | 35.11 | 37.52 | -2.41 | 37.04 | 38.86 | -1.82 | 50.92 | 49.93 | 0.99 | 54.72 | 54.42 | 0.30 | 35.49 | 37.99 | -2.50 | 51.07 | 50.95 | 0.12 |
| 4o | 68.43 | 67.65 | 0.78 | 68.08 | 67.36 | 0.72 | 60.40 | 59.08 | 1.32 | 64.87 | 63.36 | 1.51 | 69.67 | 68.57 | 1.10 | 65.82 | 64.44 | 1.38 |
| **Visual7W** Q-7B | 71.27 | 75.72 | -4.45 | 69.24 | 70.88 | -1.64 | 64.91 | 67.42 | -2.51 | 63.20 | 65.24 | -2.04 | 71.03 | 75.43 | -4.40 | 68.52 | 71.38 | -2.86 |
| DS | 74.70 | 76.63 | -1.93 | 68.30 | 70.23 | -1.93 | 56.91 | 57.80 | -0.89 | 61.92 | 64.18 | -2.26 | 74.23 | 76.30 | -2.07 | 68.43 | 69.52 | -1.09 |
| 4o | 73.22 | 76.00 | -2.78 | 73.56 | 74.67 | -1.11 | 70.78 | 71.60 | -0.82 | 67.11 | 68.78 | -1.67 | 73.06 | 75.99 | -2.93 | 72.13 | 73.19 | -1.06 |
| **Nocaps** Q-7B | 84.56 | 80.44 | 4.12 | 82.00 | 77.30 | 4.70 | 83.67 | 80.70 | 2.97 | 80.45 | 76.60 | 3.85 | 84.43 | 80.30 | 4.13 | 84.28 | 80.36 | 3.92 |
| DS | 79.22 | 75.64 | 3.58 | 70.65 | 66.87 | 3.78 | 59.17 | 53.52 | 5.65 | 65.78 | 60.84 | 4.94 | 78.92 | 75.30 | 3.62 | 71.22 | 66.84 | 4.38 |
| 4o | 85.95 | 82.66 | 3.29 | 78.22 | 71.90 | 6.32 | 85.75 | 83.10 | 2.65 | 82.04 | 77.78 | 4.26 | 85.99 | 82.70 | 3.29 | 86.07 | 83.30 | 2.77 |
| **Mix** Q-7B | 58.41 | 58.81 | -0.40 | 62.73 | 62.79 | -0.06 | 58.24 | 58.51 | -0.27 | 64.44 | 64.41 | 0.03 | 58.41 | 58.68 | -0.27 | **64.56** | **64.93** | -0.37 |
| DS | 52.02 | 50.60 | 1.42 | 48.61 | 51.00 | -2.39 | 49.21 | 47.57 | 1.64 | 57.54 | 58.13 | -0.59 | 51.39 | 50.22 | 1.17 | **57.83** | 57.02 | 0.81 |
| 4o | 66.42 | 67.22 | -0.80 | 67.42 | 67.07 | 0.35 | 62.49 | 63.00 | -0.51 | 67.69 | 67.85 | -0.16 | 66.97 | 67.77 | -0.80 | **68.02** | 67.79 | 0.23 |

Table 10: LLM evaluation consistency. Experimented with GPT-4o and Qwen-Max.

## A.6 LLM Evaluation Consistency

We present LLM (Qwen-Max) evaluated scores in text. We address the potential concern of inconsistency among different scoring models by incorporating GPT-4o as another scoring model, in Table 10. From the "Diff." columns, the differences of the two LLM-eval metrics are mostly within 3 points (out of a range of 100).

## A.7 Failure Case Study

We give three representative cases where our planning agent fails to choose a proper mRAG strategy in Fig. 6. We summarize the analysis in each box (**Analysis**). The task model is Qwen2.5-VL-7B-Inst.

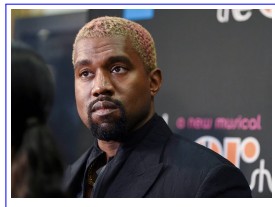

Source: Dyn-VQA (en)

**Question:** What shoe company signed he to a new multibillion dollar celebrity endorsement deal in October 2022?

**Ground truth:** He was actually dropped by many companies for offensive comments.

**Agent model:** No mRAG

**Task model:** The shoe company that signed Kanye West to a new multibillion-dollar celebrity endorsement deal in October 2022 is Adidas.

**Analysis: mRAG should be adopted to capture related news.**

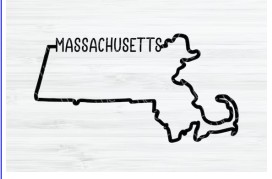

Source: Dyn-VQA (en)

**Question:** Who was the first elected female Governor of this state?

**Ground truth:** Maura Healey

**Agent model:** No mRAG

**Task model:** ...The first woman ... is Elsie Higgins.

**Analysis: mRAG should be adopted to capture related information.**

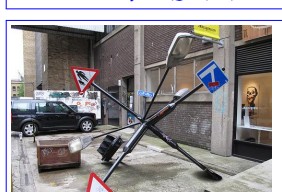

Source: Visual7W

**Question:** What is the car on?

**Ground truth:** Cement

**Agent model:** +$k_{i,t}$

**Task model:** The car appears to be parked on a paved surface, likely a street or parking area. The specific material of the pavement is not clearly visible, but it could be concrete or asphalt, which are common materials for paved surfaces.

**Analysis: Though the prediction is acceptable, this is a simple VQA query and the text/image retrievals can be saved.**

Figure 6: Three failure cases.

## A.8 HUMAN VERIFICATION OF ANNOTATION PROCESS

The automated annotation pipeline is a key factor in training data quality. We quantified the quality of our automated annotation pipeline by having three Ph.D.-level NLP researchers evaluate 100 random samples from each training set. We measured accuracy (correct annotations / total samples) for both query decomposition and correctness verification stages. Detailed results are shown in Table 11. We see that current procedures are consistent with human in most of the time. For VQAv2, we note that it is a relatively simple dataset, e.g., "*Is this a fancy supermarket?*". It usually does not involve the recognition of entities in the image content, so the rewriting of the gold query and image query is often not very meaningful. Besides, most of the time, MLLMs answer the original $q$ correctly, and performing query decomposition is actually useless (because answering original $q$ correctly is classified into category $c_1$, and the subsequent steps are omitted).

| Person 1 | Image Query Acc | Image Entity Acc | Gold Query Acc | LLM Eval Acc |
|---|---|---|---|---|
| InfoSeek | 99% | 100% | 100% | 97% |
| VQAv2 | 92% | 87% | 86% | 96% |
| WanWu | 98% | 96% | 96% | 97% |

| Person 2 | Image Query Acc | Image Entity Acc | Gold Query Acc | LLM Eval Acc |
|---|---|---|---|---|
| InfoSeek | 100% | 100% | 100% | 98% |
| VQAv2 | 94% | 89% | 90% | 98% |
| WanWu | 99% | 97% | 98% | 98% |

| Person 3 | Image Query Acc | Image Entity Acc | Gold Query Acc | LLM Eval Acc |
|---|---|---|---|---|
| InfoSeek | 99% | 100% | 100% | 99% |
| VQAv2 | 93% | 87% | 86% | 98% |
| WanWu | 99% | 96% | 97% | 98% |

Table 11: Results of human verifying the decomposition and correctness checking process.

## A.9 COMPARE WITH DEEP RESEARCH AGENT

We compared our planning agent with WebWatcher (Geng et al., 2025), with the results presented in Table 12. On the Mix dataset, which simulates a real-world situation, our method (Qwen2.5-VL-7B-Inst with the trained planning agent) outperforms WebWatcher. Furthermore, our approach achieves this with significantly lower tool-call latency, being 3× faster than WebWatcher-7B and 4.5× faster than WebWatcher-32B.

| | Mix ↑ | No mRAG | % +$k_i$ | % +$k_t$ | % +$k_{i,t}$ | % Visit | % Code | Avg. # of Rounds | Latency ↓ |
|---|---|---|---|---|---|---|---|---|---|
| **Ours (Q-7B)** | **64.93** | 24.8 | 9.5 | 43.3 | 22.3 | - | - | 1 | **4058.6** |
| **WebWatcher-7B** | 56.12 | 1.8 | 84.8 | 485.7 | - | 44.2 | 6.0 | 6.6 | 12907.2 |
| **WebWatcher-32B** | 58.92 | 0.8 | 76.0 | 1143.8 | - | 49.3 | 7.8 | 9.3 | 18740.5 |
| **Tool Latency (s)** ±std | | 0.0 | 6.4 | 1.4 | 7.8 | 21.0 ±34.1 | 0.05 | | |

Table 12: Comparison with WebWatcher-7B and 32B on Mix dataset. The latency column shows the total time spent on tool calls. The sum of % columns (of WebWatcher's) is larger than 100% because WebWatcher solves problems in a multi-round conversational manner, which may invoke multiple tool calls when answering a single VQA query.

