# OpenReview forum: "Efficient Multimodal Planning Agent for Visual Question-Answering"
_ICLR.cc/2026/Conference — Submitted to ICLR 2026_

### Official Review · Reviewer_KNJj · 2025-10-30

**Soundness:** 3
**Presentation:** 4
**Contribution:** 3
**Rating:** 4
**Confidence:** 5

**Summary:**

This paper addresses the inefficiency of static multi-stage pipelines in multimodal Retrieval-Augmented Generation (mRAG) for Visual Question-Answering (VQA)—a task requiring integration of visual and textual information. To solve this, the paper proposes a multimodal planning agent trained to dynamically decompose the mRAG pipeline: it intelligently omits unnecessary steps (e.g., skipping retrieval for simple queries) and selects only essential components (e.g., image/text retrieval for knowledge-intensive queries) based on the input VQA query.

**Strengths:**

1. Proposes a dynamic agent that optimizes mRAG pipelines for VQA, moving beyond static pipelines and prompt-based methods by integrating actionable decision-making (not just knowledge boundary detection).
2. Reduces search time by over 60% and minimizes expensive image retrieval calls compared to baselines (e.g., OmniSearch, full mRAG), addressing scalability challenges in real-world VQA.
3. Outperforms all baselines (including prompt-based methods and full mRAG) on average across six diverse VQA datasets, demonstrating that efficiency does not come at the cost of accuracy.
4.  The fine-tuned 7B-scale agent enhances performance across other MLLMs (e.g., GPT-4o, Qwen-VL-Max) without additional fine-tuning, highlighting its generalizability.

**Weaknesses:**

1. While the agent is designed to handle diverse queries, the paper does not deeply analyze how it adapts to specific query categories (e.g., knowledge-intensive vs. simple, visual vs. text-heavy). For example, it shows the agent skips mRAG for 60% of NoCaps queries but does not explain why these queries are solvable without retrieval (e.g., are they simple object-recognition questions?).
2. The paper only reports search time (time for retrieval steps) but not end-to-end inference latency (including MLLM generation time). In real-world VQA, generation time often dominates latency—especially if the agent reduces input sequence length (by skipping retrieval, thus shortening context). Without this measurement, the paper’s efficiency claims are incomplete.
3. The training data relies on three datasets: InfoSeek, VQAv2.0 (general VQA), and WanWu (Chinese VQA). There is no evaluation on domain-specific VQA (e.g., medical VQA, satellite image VQA), where mRAG is often critical (e.g., retrieving medical knowledge for image-based diagnosis). This limits the paper’s claim of the agent’s "adaptability in open-domain VQA."
4. The innovation of the paper is limited, as many engineering practices have adopted similar approaches. The method suffers from a significant scaffolding issue. Additionally, compared to Vision-Language Deep Research Agents, what are the differences in end-to-end performance and efficiency for this router-based method?

**Questions:**

See the weaknesses.

---

> ### Author Response · Authors · 2025-11-20
>
> Thanks for your detailed and in-depth reviews. Regarding your concerns
>
> > *Weakness 1: specific query categories*
>
> We included a brief introduction of each test set in Sec. 4.1.3, and how much effect mRAG has on them in Table 2. According to your suggestions, we will add
> 1. deeper analysis about how the trained agent adapts to specific query categories
> 2. typical test cases where queries are solvable with or without retrieval.
>
> Here we first give a brief analysis:
>
>
> |     Test Dataset    |     Example                                                                                                     |     Agent behavior illustration   (referring to Table 3)                                                                                                                                                                                 |
> |---------------------|-----------------------------------------------------------------------------------------------------------------|------------------------------------------------------------------------------------------------------------------------------------------------------------------------------------------------------------------------------------------|
> |     NoCaps          |     [Image]: a boat is tied to a tree with rope.       [Query]: Could you caption the content in the image.               |     Our agent decides to not use mRAG in 58.8%   of the time, and never (0%) to search the image.                                                                                                                                        |
> |     Private VQA     |     [image]: a plant that is kind of rarely   seen.    [Query]: what is the name of the plant.     |     Our agent decides to search image for   further information in 22.8% of the time and search information from both   image and textual sides in 30.9% of the time.                                                                    |
> |     Dyn-VQA (en)    |     [image]: a famous actor.     [Query]: how many children the   person has in the figure.          |     Our agent predicts that task model is able   to recognize the person, but it needs to rewrite text query for a text-search   (62.2% $+k_t$) to solve the problem (the number of children this person has   may change over time).    |
>
> > *W2: end-to-end latency*
>
> The end-to-end wall-clock time will hide the core method’s efficiency.
> The primary missing time of our current experiments contains two parts
> 1. **Base model’s inference time.** For a fair comparison with the strongest variant of *OmniSearch*, we follow *OmniSearch* and use GPT-4 as base model. However, the stability of the API is an important yet uncontrollable factor. To ensure reproducibility, we have to omit this part of the timing caused by API fluctuations. Moreover, due to the *OmniSearch*’s prompt design, both the output sequence length and the number of conversational rounds are significantly greater than those in our method. Including the GPT-4 API call time would obscure the efficiency of the core methodological contributions.
> 2. **Planning agent inference latency.** We note that this process is lightweight: the input length is short, and the output is typically a single token (see Appendix A.2). Consequently, the latency is marginal (A100-SXM-80G single GPU, 1.65s/sample). **We have already added quantitative results for this component into Sec. 5.1. Table 4 and related statements are updated in the paper (marked blue).**
>
> Lastly, we would like to note that the overall end-to-end wall-clock time of *OmniSearch* is approximately **3-6 times to ours**, taking all factors into consideration (*OmniSearch* introduces more rounds of conversations, makes more calls to GPT-4, and outputs longer sequence length, etc.).
>
> > *W3: domain-specific VQA & adaptability claim*
>
> We have incorporated test sets from various domains, including
> - knowledge-intensive (Life VQA: knowledge from daily life; Private VQA: knowledge that contains fauna, flora, architectural structures, etc.)
> - non-knowledge-intensive (NoCaps, Visual7W)
> - queries with time-sensitive answers (Dyn-VQA).
>
> The claim “improves adaptability in open-domain VQA scenarios” that we make is in the Related Work Section. It refers to our improvement in adaptability (Line 125) compared to previous work by integrating actionable decision-making into the retrieval process.

---

> ### Author Response · Authors · 2025-11-20
>
> Regarding your concern
>
> > *W4: innovation/engineering practices adopted similar approaches*
>
> Our contribution lies in
> 1. the automatic data annotation process and the fine-tuning of a planning agent.
> 2. experiments that are conducted to verify the effectiveness and efficiency of the trained agent under non-overlapping and various test datasets.
> To our best knowledge, current engineering practices mostly adopt a prompt-based method to enable retrieval (tool-call) steps [[1]](https://arxiv.org/pdf/2411.02937).
> We would be grateful for further clarification about similar *engineering practices*.
>
> > *W4: compared to Vision-Language Deep Research Agents*
>
> Our method proposes a 7B-level multimodal planning agent, aiming at mitigating the inefficiency limitations while maintaining task performance. A stronger task model leads to better performance by nature. We have experimented with GPT-4o (which is a powerful MLLM) in Sec. 5.2, and find our 7B-level agent still helpful.
> Could you please give more specific settings of *“comparing to VL Deep Research Agents”* (e.g., which VL DR Agent you would like to see the comparison)? We are more than willing to supplement such an experiment.

---

> ### Author Response · Authors · 2025-11-27
>
> Thank you again for your valuable review of our paper. We are writing to gently follow up on our rebuttal submitted one week ago.
>
> To address your key concerns, we have made the following updates:
>
> - **Specific query categories:** We provided a detailed analysis of how our agent adapts to specific query categories (W1).
>
> - **End-to-End Latency:** Quantitative results for our planning agent's latency -> Section 5.1, Table 4 (**marked blue** for your convenience)
>
> Furthermore, we are experimenting with adding the performance of Qwen3-VL-Thinking (with search tools) and will incorporate the results as soon as possible.
>
> We would be very grateful if you could take a moment to see if our responses and planned revisions have adequately addressed your concerns. Your feedback would be invaluable to us.

---

> ### Comment · Reviewer_KNJj · 2025-11-28
> **Thanks for your reponse.**
>
> 1. Please compare your method with VL DR Agents such as WebWatcher (as described in "Breaking New Frontier of Vision-Language Deep Research Agent").
> 2. What are the test results of your method on the MMSEARCH-PLUS benchmark?

---

> > ### Author Response · Authors · 2025-11-28
> >
> > We appreciate your responses. We are launching the two experiments.

---

> > ### Author Response · Authors · 2025-12-02
> >
> > 1.	**The results of WebWatcher are shown in table below (also added to paper Sec. A9, marked blue)**. We incorporate 4 tools to WebWatcher, i.e., text search, image search, visit (to websites) and Python interpreter, to showcase its performance. The summation of % columns of WebWatcher is beyond 100% because WebWatcher solves problems conversationally and it might invoke tools more than once in solving each VQA query. We note that the time latency WebWatcher introduces is **3-4.5 times** to ours, showing the inefficiency of such Deep Research Agents compared with our method.
> >
> >     | | Mix ↑ | No mRAG | % +$k_i$ | % +$k_t$ | % +$k_{i,t}$ | % Visit | % Code | Avg. # of Rounds | Latency ↓ |
> >     | :--- | :---: | :---: | :---: | :---: | :---: | :---: | :---: | :---: | :---: |
> >     | Ours (Q-7B) | **64.93** | 24.8 | 9.5 | 43.3 | 22.3 | - | - | 1 | **4058.6** |
> >     | WebWatcher-7B | 56.12 | 1.8 | 84.8 | 485.7 | - | 44.2 | 6.0 | 6.6 | 12907.2 |
> >     | WebWatcher-32B | 58.92 | 0.8 | 76.0 | 1143.8 | - | 49.3 | 7.8 | 9.3 | 18740.5 |
> >     | Tool Latency (s) ±std | | 0.0 | 6.4 | 1.4 | 7.8 | 21.0 ±34.1 | 0.05 | | |
> >
> >
> > 2.	As for the *”MMSEARCH-PLUS benchmark”*, we would like to point out a potential bug in the official release of the [MMSearch-Plus dataset](https://huggingface.co/datasets/Cie1/MMSearch-Plus). We have tried:
> >
> >   a)	Download and load from disk. Its metadata file, [state.json](https://huggingface.co/datasets/Cie1/MMSearch-Plus/blob/main/state.json), indicates there are **two** arrow-formatted data files, the repository actually contains **five**.
> >
> >   b)	Load from online. Error log:
> >
> >   ```bash
> >   Downloading data files: 100%...
> >   …
> >   {'question': Value(dtype='string', id=None), 'answer': Sequence(feature=Value(dtype='string', id=None), length=-1, id=None),  'num_images': Value(dtype='int64', id=None), 'arxiv_id': Value(dtype='string', id=None), 'video_url': Value(dtype='string', id=None), 'category': Value(dtype='string', id=None), 'difficulty': Value(dtype='string', id=None), 'subtask': Value(dtype='string', id=None), 'img_1': Image(decode=True, id=None), 'img_2': Image(decode=True, id=None), 'img_3': Image(decode=True, id=None), 'img_4': Image(decode=True, id=None), 'img_5': Image(decode=True, id=None)}
> >   because column names don't match
> >   …
> >   datasets.builder.DatasetGenerationError: An error occurred while generating the dataset
> >   ```
> > As of our submission, we have not found an official solution or clarification for this inconsistency. To adhere to the anonymous review policy, we were unable to raise a public issue on the repository to seek resolution.
> >
> >
> > ----
> >
> > Lastly, we would like to note that, according to the [ICLR26 Reviewer Guide](https://iclr.cc/Conferences/2026/ReviewerGuide)
> >
> > 1.	Authors are not required to cite and compare with very recent works, which means papers published after July 24, 2025.
> >
> > 2.	*”Authors are not required to compare to papers solely on arXiv.”*
> >
> > However, WebWatcher and MMSearch-Plus are both released **after July 24, 2025** on **arXiv**.
> >
> > The guide explicitly states that *”The lack of such comparisons cannot be a basis for rejection”*.

---

### Official Review · Reviewer_UG5S · 2025-10-30

**Soundness:** 3
**Presentation:** 3
**Contribution:** 2
**Rating:** 4
**Confidence:** 4

**Summary:**

The paper’s core idea, learning to pick among four retrieval/actions to avoid overusing multimodal RAG, is timely and practically useful, and the empirical comparison to a strong prompt-based controller with ~60% search-time reduction is attractive. The evaluation is also commendably broad across six datasets, and the fact that the learned planner transfers to other MLLMs suggests the approach is not overfit to one model. However, a few high-impact weaknesses remain: (i) the whole pipeline leans on unquantified LLM-generated labels, with no noise/quality reporting (W1); (ii) the main effectiveness numbers use LLM-based scoring instead of standard VQA metrics or human evaluation, so the gains are hard to situate in the VQA literature; (iii) efficiency is reported via estimated tool-time, not end-to-end latency, so the headline 60% saving is not fully substantiated; and (iv) there is no comparison to other learned/routing agents, so it is unclear how much is due to the specific proposed planner vs. “any learned router”, combined with rebalanced training that may not reflect real-world distributions, and a lack of significance reporting.

**Strengths:**

S1: Targets a concrete, high-impact pain point in multimodal RAG for VQA, unnecessary image/text retrieval and inflated context length, by turning the pipeline into an adaptive one.

S2: Automated LLM-based data construction (visual query decomposition + correctness checking) enables building large supervision without heavy manual labeling.

S3: Broad evaluation on six heterogeneous VQA(-like) datasets, demonstrating generality across dynamic, knowledge-intensive, and easier visual tasks.

S4: Strong empirical gain over the prompt-based OmniSearch controller: similar or better answer quality with about 60%–66% lower search time.

S5: Shows cross-model transfer to other MLLMs (GPT-4o, Qwen-VL-Max, DeepSeek-VL), suggesting the learned policy is not tied to a single backbone.

**Weaknesses:**

W1: Despite the automated LLM-based data pipeline being a key enabler, the approach still relies heavily on LLM-generated supervision for both decomposition and correctness checking, but the paper does not quantify annotation noise or its impact.

W2: Although the evaluation covers six datasets and looks broad, it leans on LLM-based scoring rather than standard VQA metrics or human judgment, which weakens comparability to prior VQA work.

W3: While the method reports a 60%+ search-time reduction over a strong prompt-based controller, this efficiency is derived from estimated tool/API times rather than end-to-end wall-clock measurements on a fixed setup.

W4: The four routing categories are well motivated and interpretable, but the training set is rebalanced across them, so the learned policy is not evaluated under the real, potentially skewed, test-time distribution.

W5: The system still depends on an inference-time query-rewriting MLLM; if that component changes or is weaker in deployment, some of the claimed savings/accuracy may erode.

W6: There is no analysis of misrouted or failed cases (e.g., choosing no mRAG when external knowledge was needed), so failure modes remain opaque.

W7: The pipeline uses multiple Qwen variants/APIs for different substeps, which could hurt replicability when those endpoints change.

**Questions:**

Q1: Since you rely on LLM-based scoring, add at least one standard VQA-style metric (e.g., VQAv2 accuracy / EM) on the English datasets and report variance or significance for the LLM-eval scores, so we can judge robustness.

Q2: Please quantify the quality of the LLM-generated supervision: report error/noise rates for visual query decomposition and correctness checking on a manually annotated subset, and add an ablation showing planner accuracy with/without cleaned labels.

---

> ### Author Response · Authors · 2025-11-20
>
> Thanks for your in-depth and detailed review. Regarding your concern
>
> > *W1: unqualified LLM-generated supervision & Q2: error/noise rates for visual query decomposition and correctness checking*
>
> The automated annotation pipeline is indeed a key factor in training data quality. We quantified the quality of our automated annotation pipeline by having three Ph.D.-level NLP researchers evaluate 100 samples from each training set. We measured accuracy (correct annotations / total samples) for both decomposition and correctness verification stages. (You can also refer to the updated paper Sec. A.8)
>
> | person 1 | image query acc | image entity acc | gold query acc | LLM eval acc |
> |----------|-----------------|------------------|----------------|--------------|
> | infoseek | 99%             | 100%             | 100%           | 97%          |
> | vqav2    | 92%             | 87%              | 86%            | 96%          |
> | wanwu    | 98%             | 96%              | 96%            | 97%          |
> Table 1
>
> | person 2 | image query acc | image entity acc | gold query acc | LLM eval acc |
> |---|----|-----|-----|-----|
> | infoseek | 100%            | 100%             | 100%           | 98%          |
> | vqav2    | 94%             | 89%              | 90%            | 98%          |
> | wanwu    | 99%             | 97%              | 98%            | 98%          |
> Table 2
>
> | person 3 | image query acc | image entity acc | gold query acc | LLM eval acc |
> |-----|-----|-------|-------|------|
> | infoseek | 99%             | 100%             | 100%           | 99%          |
> | vqav2    | 93%             | 87%              | 86%            | 98%          |
> | wanwu    | 99%             | 96%              | 97%            | 98%          |
> Table 3
>
> Our evaluation demonstrates high reliability for both LLM-based query decomposition and correctness verification. Furthermore, the consistent performance improvements across multiple benchmarks (Tables 3 and 5 in the paper) empirically validate our annotation quality.
>
> > *W2: LLM-based scoring & Q1: VQA-style metric*
>
> We reported **token accuracy** in Lines 298-299 and detailed results in Sec. A.5, which aligns with LLM-based scoring. Additionally, our human evaluation (Tables 1-3 above) validates the consistency between LLM-based scoring and human judgment.
>
> > *W3: end-to-end wall-clock*
>
> The end-to-end wall-clock time will hide the core method’s efficiency.
> The primary missing time of our current experiments contains two parts
> 1. **Base model’s inference time.** For a fair comparison with the strongest variant of *OmniSearch*, we follow *OmniSearch* and use GPT-4 as base model. However, the stability of the API is an important yet uncontrollable factor. To ensure reproducibility, we have to omit this part of the timing caused by API fluctuations. Moreover, due to the *OmniSearch*’s prompt design, both the output sequence length and the number of conversational rounds are significantly greater than those in our method. Including the GPT-4 API call time would obscure the efficiency of the core methodological contributions.
> 2. **Planning agent inference latency.** We note that this process is lightweight: the input length is short, and the output is typically a single token (see Appendix A.2). Consequently, the latency is marginal (A100-SXM-80G single GPU, 1.65s/sample). **We have already added quantitative results for this component into Sec. 5.1. Table 4 and related statements are updated in the paper (marked blue).**
>
> Lastly, we would like to note that the overall end-to-end wall-clock time of *OmniSearch* is approximately 3-6 times to ours, taking all factors into consideration (*OmniSearch* introduces more rounds of conversations, makes more calls to GPT-4, and outputs longer sequence length, etc.).
>
> > *W4: rebalancing…real distribution*
>
> The rebalancing is to stabilize training and make sure each category receives enough training supervision. A skewed training set usually needs further tricks to stabilize training, empirically. We would like to point out that WanWu is collected from real-world distribution. Thanks for pointing this out and we will add the clarification to the paper. Adding InfoSeek and VQAv2.0 mainly aims to incorporate more domains of VQA queries and ensure generalization performance.
>
> We also would like to clarify that our training and test sets are kept completely non-overlapping (Line 250). Our "Mix" data was designed to simulate a general real-world scenario by creating a uniform distribution, with an equal number of samples from all constituent test sets. This ensures a fair and balanced evaluation of the model's generalization across different domains. If the reviewer's concern is that this uniform distribution does not reflect a specific real-world skew, we would be grateful for further suggestions on what a more representative, non-skewed distribution would look like in this context.

---

> > ### Author Response · Authors · 2025-11-20
> >
> > > *W5: query-rewriting MLLM*
> >
> > Thanks for noting this important practical consideration. We adopt a uniform query-rewriting model to **ensure core method’s consistency** (Line 268). Specifically, this is to fairly verify the effectiveness of our method on both original sampled MLLM (Sec. 4.2) and other MLLMs variants (Sec. 5.2). Thus, the planning agent’s ability to choose correct retrieval can be isolated from the query-rewriting step.
> >
> > > *W6: failed cases*
> >
> > We have added a Failure Case Study to the paper, Sec. A.7, e.g., where mRAG should be adopted but the router decides not to (marked blue). Here are the 3 cases:
> >
> > **Failed case 1** (Dyn-VQA en):
> > - Image: [A picture of Kanye West].
> > - Question: What shoe company signed he to a new multibillion dollar celebrity endorsement deal in October 2022?
> > - Ground truth answer: He was actually dropped by many companies for offensive comments.
> > - Agent model’s prediction: No mRAG
> > - Task model’s incorrect output: The shoe company that signed Kanye West to a new multibillion-dollar celebrity endorsement deal in October 2022 is Adidas.
> > - **Case analysis: mRAG should be adopted to capture related news.**
> >
> > **Failed case 2** (Dyn-VQA en):
> >   - Image: [A map of Massachusetts]
> >   - Question: Who was the first elected female Governor of this state?
> >   - Ground truth answer: Maura Healey
> >   - Agent model’s prediction: No mRAG
> >   - Task model’s incorrect output: …The first woman … is Elsie Higgins.
> >   - **Case analysis: A Google (text) search can reveal that Maura Healey is the expected answer.**
> >
> > **Case 3: room for improvement.** (Visual7W)
> >   - Image: [A car on Cement, with a fallen street lamp]
> >   - Question: What is the car on?
> >   - Ground truth answer: Cement
> >   - Agent model’s prediction: $+k_{i,t}$
> >   - Task model’s output: The car appears to be parked on a paved surface, likely a street or parking area. The specific material of the pavement is not clearly visible, but it could be concrete or asphalt, which are common materials for paved surfaces.
> >   - **Case analysis: Though the prediction is acceptable, this is a simple VQA query and the text/image retrievals can be saved.**
> >
> >
> > > *W7: endpoints changing hurts replicability*
> >
> > The Qwen variants/APIs we utilize are stable. We will release the implementation details alongside the code, including all API calling procedures.
> >
> > We have updated our paper according to your suggestions, and the updated parts are colored in blue.

---

> ### Author Response · Authors · 2025-11-27
>
> Thank you again for your valuable review of our paper. We are writing to gently follow up on our rebuttal submitted last week.
>
> To address your key concerns, we have updated the paper with new experiments and analyses:
>
> - **Data Quality Validation:** A new human evaluation study, now presented in Appendix A.8.
>
> - **Efficiency Analysis:** Quantitative latency results updated for our lightweight planning agent in Section 5.1, Table 4.
>
> - **Failure Analysis:** Include a new failure case analysis in Appendix A.7.
>
> All revisions in the updated paper are marked in blue for your convenience. We would be very grateful if you could take a moment to see if our revisions have adequately addressed your concerns. Your feedback is invaluable to us.

---

> > ### Comment · Reviewer_UG5S · 2025-11-27
> >
> > I have read the responses, and I would like to thank the authors for the hard work in the rebuttal! I think most of my concerns are addressed, and if the answers are integrated into the paper, it will make it a strong paper ready for acceptance. I will bump up my scores accordingly.

---

### Official Review · Reviewer_zRP3 · 2025-10-31

**Soundness:** 2
**Presentation:** 2
**Contribution:** 2
**Rating:** 4
**Confidence:** 4

**Summary:**

This paper addresses Visual Question Answering (VQA) by proposing a multimodal planning agent to optimize multimodal Retrieval-Augmented Generation (mRAG) workflows. The key idea is to dynamically decide whether to perform image retrieval, text retrieval, both, or neither, rather than executing all steps in a fixed pipeline. The authors train this agent through automatic data annotation and fine-tuning, and evaluate it across multiple VQA datasets.

**Strengths:**

- The paper identifies a real problem: existing mRAG pipelines suffer from multi-stage dependencies and redundant computations. This is a well-motivated research direction.
- The method is essentially a classifier that predicts which retrieval path to take for each VQA input. This makes it easy to implement and integrate into existing systems. The automatic data generation process also avoids expensive manual annotation.
- The experiments results show significant reduction in retrieval operations and search time while maintaining comparable task performance across multiple datasets.

**Weaknesses:**

- My main concern is the significance of the contribution. The proposed method is essentially training an LLM-based classifier for a four-way classification task, rather than a true multi-step planning agent. While effective in practice, the methodological innovation is relatively limited.
- The evaluation scope is somewhat narrow: 1) The main results rely solely on LLM Eval scores (0-100) without standard VQA metrics like accuracy, BLEU, or ROUGE. 2) Only Qwen-Max is used for evaluation, and the evaluation protocol (prompts, rubrics) is not fully transparent.
- The paper does not compare against specialized mRAG optimization methods or recent VQA SOTA systems (such as state-of-the-art agent-based VQA approaches). The comparisons feel more like ablations rather than positioning against the broader literature.
- Training labels are entirely generated by automated models without reported quality checks or human verification rates.
- All scores are single values without error bars, confidence intervals, or significance tests.

**Questions:**

The paper describes the main contribution as a "planning agent," but the system only makes a single four-way classification decision rather than multi-step planning. Could you explain why you call it a "planning agent" instead of a "strategy classifier"? Do you plan to extend it to multi-step, context-aware decision-making in future work?

---

> ### Author Response · Authors · 2025-11-20
>
> Thanks for your detailed reviews. Regarding your concern
>
> > *significance of the contribution*
>
> Our contribution lies in
> 1. the automatic data annotation process.
> 2. experiments that are conducted to verify the effectiveness and efficiency of the trained agent under non-overlapping and various test datasets.
> To the best of our knowledge, there are no related works that attempt to train such an agent using a similar data annotation process.
>
> > *evaluation scope*
>
> 1) We mentioned the **token accuracy** metric in Line 298-299 of the text and reported the detailed number in Sec. A.5.
> 2) We have added another LLM for evaluation (GPT-4o with the same prompts) and added it to the paper, Sec. A.6. We present the results on three representative models on **all test sets**. The evaluation protocol references [Llama-Index](https://github.com/run-llama/llama_index/blob/a719cee8ba43bb3b6c2e3b00d2458a22a7583e88/llama-index-core/llama_index/core/evaluation/correctness.py#L19). We have added the prompts and original reference to the paper, Sec. A.1.
>
> Table Title: GPT-4o evaluation comparison with Qwen-Max
>
> [Anonymous TABLE](https://ibb.co/9krmMqph)
>
> From the “Diff.” columns, we can see that the differences of the two LLM-eval metrics are mostly within 3 points (out of a range of 100). Furthermore, we even obtain better GPT-4o evaluated performances (Mix row, Ours column, three **bold (best)** numbers).
>
> > *compare against recent VQA SOTA systems*
>
> *OmniSearch* (with GPT-4 as the base model) is a SOTA VQA agent as stated in its [ICLR25 paper](https://arxiv.org/pdf/2411.02937), Sec. 5.2 the first line of (1), on the Dyn VQA dataset. We show that our agent outperforms *OmniSearch* on the same setting (with GPT-4 as the base model) in Table 4. Even so, we are willing to address your concern by providing additional results if you could specify which “VQA SOTA systems” you would like us to compare against.
>
> > *training labels w/o human verification*
>
> Verifying the training labels manually is actually undesirable because humans’ judgment might not align with the actual case. Whether a retrieval will be helpful should depend on (M)LLMs themselves, of which humans may not have sufficient knowledge. For example,
> 1. given a VQA query asking about a piece of news in the recent past, it is hard to judge by humans whether the relevant information is included in the pre-training corpus or post-training stages.
> 2. given a knowledge-intensive VQA query asking the species of a plant, though it is likely to appear in a common pre-training corpus like Wikipedia, models are still likely to hallucinate [[NAACL24]](https://arxiv.org/abs/2403.20009v1).
>
> However, we performed a manual verification on the **prerequisite steps (rather than the final labels)**- the “query decomposition” and “correctness checking” steps-by having three Ph.D.-level NLP researchers evaluate 100 samples from each training set. We measured accuracy (correct annotations / total samples) for both decomposition and correctness verification stages. (Please refer to the updated paper Sec. A.8)
>
> > *error bars, confidence interval*
>
> The variance of our method mainly comes from the planning agent’s decision and task model.
> Current experiment results can be seen as Pass@1. We will utilize our computational resources to the fullest extent to add significance tests.
>
> > *Questions: multi-step agent/planning agent naming*
>
> 1. We are considering extending the method to multiple (inference) step, e.g., multi-round text/image search in a conversational manner for more complex cases. But note that our current involved test sets are mostly solvable by procedures in Figure 1.
> 2. Our use of the term "planning agent" is primarily motivated by the fact that the agent must make decisions across multiple action spaces.
>
> We have updated our paper according to your suggestions and the updated parts are colored in blue.

---

> ### Author Response · Authors · 2025-11-27
>
> Thank you again for your valuable review of our paper. We are writing to gently follow up on our rebuttal submitted one week ago.
>
> To address your key concerns, we have made the following updates to the paper:
>
> - **Expanded Evaluation:** GPT-4o as an additional LLM evaluator -> Appendix A.6.
>
> - **Verification of Annotation Pipeline:** A manual verification study by three Ph.D. researchers -> detailed in Appendix A.8.
>
> Furthermore, we are currently running additional experiments to provide **significance tests** for our results, as you suggested, and will incorporate them as soon as they are available.
>
> All revisions in the updated paper are marked in blue for your convenience. We would be very grateful if you could take a moment to see if new results have adequately addressed your concerns. Your feedback would be invaluable to us.

---

### Official Review · Reviewer_1Q3V · 2025-11-10

**Soundness:** 3
**Presentation:** 3
**Contribution:** 3
**Rating:** 8
**Confidence:** 4

**Summary:**

This paper introduces a multimodal planning agent designed to optimize the efficiency of Retrieval-Augmented Generation (mRAG) for Visual Question-Answering (VQA) tasks. Existing mRAG approaches often utilize rigid, multi-stage pipelines (e.g., always performing image retrieval, then query rewriting, then text retrieval), which can be computationally expensive, slow, and sometimes redundant for simpler queries. The authors propose a dynamic approach where a fine-tuned 7B agent (Qwen2.5-VL-7B-Inst) analyzes the input VQA query and determines the necessary mRAG steps: no RAG, image RAG only, text RAG only, or both.

The agent trains by using an automated data construction pipeline that employs a larger, stronger MLLM (Qwen2.5-VL-72B) in order to generate "gold queries" and determine which ground truth labels for which retrieval steps are actually required to correctly answer a given training question.

Empirical results across six diverse VQA datasets demonstrate that this agent-based approach reduces search time by over 60% compared to prompt-based baselines (OmniSearch) and significantly cuts down on expensive image retrieval calls while maintaining or slightly improving overall VQA accuracy. The paper also shows that this planning agent can effectively guide other downstream MLLMs (e.g., GPT-4o, DeepSeek-VL) without further fine-tuning.

**Strengths:**

Practical Significance: Addressing the inefficiency of rigid RAG pipelines is a highly relevant problem for real-world deployment of MLLMs. The reported >60% reduction in search time is a substantial practical improvement.

Strong Empirical Results: The approach strikes a balance of efficiency with strong performances. It manages to match or outperform the computationally expensive "$+k_{i,t}$" full RAG baseline, while using a fraction of the retrieval calls.

Transferability: A strength of the agent is that it can improve the performance of other, closed-source MLLMs (such as GPT-4o) when used as a plug-and-play planner. An example is shown in Table 5. This increases the general utility of the proposed method.

Comprehensive Evaluation: The use of six diverse datasets, ranging from the standard benchmarks of Visual7W to the more complex and dynamic DynVQA, provides a robust assessment of the agent's adaptability.

**Weaknesses:**

Dependency on Oracle for Training Data: Automated data annotation has a strong dependency on establishing if a model can answer a question with/without RAG. This may be inherently noisy if the base model used for annotation itself has no clearly demarcated performance boundaries. The authors generate the gold query with a strong model, Qwen-72B, but perhaps biases in that model's knowledge boundary propagate to the agent.

Comparison to Baseline for "Agents" is Limited: The main comparison to dynamically generated approaches is to OmniSearch, a prompt-based approach. Comparisons to other recent adaptive RAG approaches (albeit unimodal, adapted for the task) would better support the claim of outperforming existing planning approaches.

**Questions:**

1. Latency Overhead: While the search time is reduced, what is the computational overhead of introducing the extra agent step (inference of the 7B model) before the main solver? Is this negligible compared to the retrieval latencies listed in Section 5.1?

2.  Sensitivity to Annotation Model: How sensitive is the final agent's performance to the choice of the MLLM used for the automated data annotation phase - currently Qwen2.5-VL-72B? Would using a weaker model for annotation significantly degrade the planner's judgment?

3.  Understanding LoRA rank: In Section 5.3, you mention that LoRA (r=8) failed while (r=32) succeeded. Do you have any guesses as to why rank 8 was not sufficient for this task, since often in fine-tuning of NLP tasks, r=8 is sufficient? Does this suggest that the routing task requires capturing more complex relationships than typical instruction-following?

4. Training Data Balance: Table 1 is a representative balance of the final training data categories, such as 30k for No mRAG, Query mRAG, Both mRAG, but only 8.8k for Image mRAG, and so on until the very last. Was this imbalance intentional, based on natural distribution? And did this affect the agent's willingness to select specifically "Image mRAG"?

---

> ### Author Response · Authors · 2025-11-20
>
> Thanks for your detailed reviews. Regarding your concern
>
>
> > *generate the gold query with a strong model…biases…*
>
> We have added an experiment to quantify the human-level quality of both “decomposition” and “correctness checking” processes. Please refer to the updated paper Sec. A.8. It can be seen that the LLM-based “decomposition” and “correctness checking” processes are reliable.
>
> > *Comparison to agents*
>
> [OmniSearch](https://arxiv.org/pdf/2411.02937) is a SOTA agent (on Dyn-VQA). We are considering adding comparisons with recent RAG/Agent approaches within our computational budget. To ensure we address your concerns appropriately, could you kindly recommend specific RAG/Agent-based methods that would be most relevant for comparison?
>
> > *Q1: Latency Overhead*
>
> Yes, the inference latency is negligible because the number of output token is basically one (Sec. App A.2). Thanks for noting this and we have added quantitative results for this component to Sec. 5.1 (marked blue).
>
> > *Q2: Sensitivity to Annotation Model*
>
> We would like to clarify that we actually used Qwen2.5-VL-7B (not the 72B version) for the automated data annotation phase, as stated in Line 231 ($M_\theta$). This is a relatively modest-sized model compared to state-of-the-art MLLMs. If you consider the 7B model still too strong for this analysis, we would be happy to conduct an additional ablation study using a weaker annotation model.
>
>
> > *Q3: LoRA rank*
>
> Yes, we believe this suggests that the routing task requires capturing more complex relationships than typical instruction-following. Potential Reason:
> 1. This routing task is rarely seen during the pre-training stage because pre-training corpus is mostly coherent texts instead of action-included contents like “which retrieval to take (or, whether to take)”.
> 2. The task’s gold output depends on the VQA queries’ domain and MLLMs’ internal knowledge. Besides, the involved domains are huge.
>
> > *Q4: Training Data Balance*
>
> The current imbalance is not intentional based on natural distribution. The reason for the smaller proportion of Image mRAG training data is that our annotated training set contains only this amount. We controlled the other categories at 30k to prevent an excessive imbalance compared to the Image mRAG category, while ensuring sufficient supervision signals for each category. Some of our preliminary experiments showed that if there is a severe imbalance among various categories, the model may easily hack the training data and ignore those categories with much lower proportions.

---

### Public Comment · ~Zhuo_Chen16 · 2026-03-15
**Summary of Rebuttal**

TL; DR

The overall ratings of our paper, **before the OpenReview API Security Incident**, are

Reviewer 1Q3V: **8**

Reviewer zRP3: **4** -No reply from zRP3 in rebuttal phase at all.

Reviewer UG5S: **8** -Raised scores before the incident. ([History evidence](https://openreview.net/revisions?id=f5qkH8EaFc), [screenshot evidence](https://ibb.co/4gJfG48y))

Reviewer KNJj: **4** -Requested unnecessary experiments after the incident. OpenReview closed the subsequent discussion.

**AC 5p2H's meta-review contains severe factual hallucinations**. It claims concerns were unaddressed when they were in fact fully rebutted, and evidence ([a GPTZero analysis](https://ibb.co/d0jY9x0K)) indicates a high probability of it being AI-generated

----

We received 2 reviewers’ replies (out of 4 reviewers) in rebuttal. Below is a summary.

1. **Significant Score Increase**. Reviewer UG5S raised the score from 4 to 8 after reading our rebuttal (See his/her response “…make it a strong paper ready for acceptance. I will bump up my scores accordingly.”). The timing is before the recent OpenReview API Security Incident. We have OpenReview history and an anonymized screenshot as evidence.

2. We have also addressed the follow-up comments from Reviewer KNJj (who responded after the security incident) by conducting the additional experiments they requested. **The results are presented in our follow-up reply to the reviewer and revised paper (Sec. A.9, marked blue).** We note that, according to the ICLR26 Reviewer Guide, the requested experiments do not constitute a basis for rejection.

---

### Meta-Review · Area_Chair_5p2H · 2025-12-28

**Summary:**

The paper proposes a multimodal planning agent for optimizing multimodal Retrieval-Augmented Generation (mRAG) pipelines in VQA  tasks. The agent dynamically decomposes queries into sub-tasks to reduce inefficiencies like unnecessary retrieval calls, trained via automated data annotation using a stronger MLLM

In rebuttal, authors addressed some reviewer concerns by adding experiments (e.g., human verification of annotation quality by three PhD-level NLP researchers, failure case analysis, additional LLM evaluators like GPT-4o, and latency details), and one reviewer raised the score, leaving the mixed rating (8,8,4,4).

While the work shows practical efficiency gains and broad evaluation, it leans more towards engineering optimization than academic research. This is due to a lack of theoretical depth in explaining the agent's decision-making process, and methodological innovation is limited to framing a simple four-way classifier as a "planning agent." Therefore, I recommend rejection.

**Reviewer Concerns:**

Key concerns about novelty, limited innovation (e.g., essentially a classifier rather than true multi-step planning), evaluation transparency (e.g., reliance on LLM scoring), annotation noise, and incomplete efficiency measurements (e.g., end-to-end latency) remain partially unaddressed despite rebuttal efforts.

**Reviewer Scores:**

UG5S raised the score from 4 to 8.

---

### Decision · Program_Chairs · 2026-01-26

Reject